behaviour/cognition

music, digital traces, diurnal patterns, music information retrieval, audio features

**Author for correspondence:**
Ole Adrian Heggli
e-mail: ole.heggli@clin.au.dk

# Diurnal fluctuations in musical preference

## Ole Adrian Heggli, Jan Stupacher and Peter Vuust

Center for Music in the Brain, Department of Clinical Medicine, Aarhus University and The Royal Academy of Music Aarhus/Aalborg, Aarhus, Denmark

  OAH, 0000-0002-7461-0309; JS, 0000-0002-2179-2508; PV, 0000-0002-4908-735X

The rhythm of human life is governed by diurnal cycles, as a result of endogenous circadian processes evolved to maximize biological fitness. Even complex aspects of daily life, such as affective states, exhibit systematic diurnal patterns which in turn influence behaviour. As a result, previous research has identified population-level diurnal patterns in affective preference for music. By analysing audio features from over two billion music streaming events on Spotify, we find that the music people listen to divides into five distinct time blocks corresponding to morning, afternoon, evening, night and late night/early morning. By integrating an artificial neural network with Spotify's API, we show a general awareness of diurnal preference in playlists, which is not present to the same extent for individual tracks. Our results demonstrate how music intertwines with our daily lives and highlight how even something as individual as musical preference is influenced by underlying diurnal patterns.

## Statement of relevance

Today, most music listening happens on online streaming services allowing us to listen to what we want when we want it. By analysing audio features from over two billion music streaming events, we find that the music people listen to can be divided into five different time blocks corresponding to morning, afternoon, evening, night and late night/early morning. These blocks follow the same order throughout the week, but differ in length and starting time when comparing workdays and weekends. This study provides an extremely robust and detailed understanding of our daily listening habits. It illustrates how circadian rhythms and 7-day cycles of Western life influence fluctuations in musical preference on an individual as well as population level.

## 1. Introduction

As humans our everyday life is guided by diurnal patterns. Most obviously we tend to sleep at night and to be active during the

day, but diurnal patterns also manifest in a range of biochemical and physiological systems [1,2]. These rhythms stem from natural selection favouring endogenous circadian rhythms, intrinsically locked to the planet's rotational period [3,4]. Not all these patterns follow a simple sinusoidal light–dark cycle, but instead exhibit complex behaviours such as transitional periods and time-lagged interactions [5].

In contrast to circadian rhythms the diurnal patterns of human society may be more discrete. Many of the recurring daily activities defining our everyday life, such as waking up, going outside and having meals, exhibit population-level regularities, although with diverging pathways through the day [6–10]. Over the last few decades, it has become apparent that cognitive and affective states exhibit systematic diurnal patterns too [11–13]. Many of these patterns have been found using desynchrony protocols, designed to separate the contribution of the sleep–wake cycle and circadian systems [14]. However, with desynchrony experiments often being weeks long, this puts a natural limit on the sample size it is feasible to collect. To investigate population-level diurnal patterns there is hence a need to look at alternative avenues for data collection.

A promising avenue for identifying diurnal affective patterns comes from digital traces. Our online activities leave digital traces with visible daily rhythms. Such traces can reflect usage statistics, such as found in editorial activity on Wikipedia [15], but also provide insight into affective states, such as the fluctuations in positive and negative affect in tweets [16], and in musical intensity [17]. Music is of particular interest due to its ubiquitousness, accessibility and its role in reflecting affective states [18–21]. In addition, music listening has rapidly become a predominantly digital and online activity, with the Recording Industry Association of America reporting that streaming accounted for 79% of US music industry revenue in 2019, compared to just 5% in 2009 [22].

Further motivation for using music to explore population-level diurnal patterns comes from historic evidence for a link between time of day and musical content. For instance, there are cyclic differences in music composed for the Liturgy of the Hours in Western Christianity, and Western serenades and nocturnes are often intended for performance in the evening [23]. A similar awareness of time of the day can be found in the Hindustani music tradition, wherein a raga is often intended for specific time windows of the day to maximize its emotional impact [24,25]. Contemporarily, a recent study on music streaming data from 1 million individuals finds a preference for energetic music during working hours, with a gradual shift to relaxing music late at night [17]. Hence, investigating diurnal variation in musical content could provide unique insight into population-level affective states.

Here, we use the music streaming sessions dataset (MSSD) released by Spotify to identify diurnal patterns of musical content [26]. The dataset contains over 2 billion music streaming events collected from streaming sessions defined as a period of active listening (wherein the listener did not have more than a minute of inactivity between consecutive tracks). These sessions were sampled uniformly at random from users of Spotify over an eight-week period, but demographic information and individual-level data are not included. The MSSD is one of the largest publicly available datasets on music listening behaviour including information on when, during the day, a particular track is listened to. While multiple other large-scale datasets on listening behaviour exist, such as the LFM-1b, LFM-2b and The Music Listening Histories dataset, the MSSD is unique due to containing pre-calculated musical audio features [27–29].

The streaming events in the MSSD are associated with 3.7 million unique tracks and their linked audio features. These features are derived by analysing audio files and indicate various descriptors of music content including both basic measures, such as perceived loudness and tempo, and more complex measures, such as danceability and beat strength (for a detailed overview, see Methods). Prior research within the field of music information retrieval has shown how such features relate to many perceptual and emotional aspects of music listening, such as emotional valence, tension arousal and the sensation of groove [30,31]. By leveraging the rich information in the MSSD along with two behavioural follow-up studies we show that audio features allow us to quantify distinct diurnal patterns of musical content.

## 2. Results

We first filtered the MSSD to retain only streaming events wherein most or all tracks were listened to, and where the listener performed a maximum of one seek-forward or -backward operation. We then collated streaming events for each hour of the week (with hour 1 being 00.00 on Monday and hour 168 being 23.00 on Sunday) and performed $k$-means clustering on normalized averaged data (for details, see Methods).

(a)

fluctuations in audio features clustered to five subdivisions

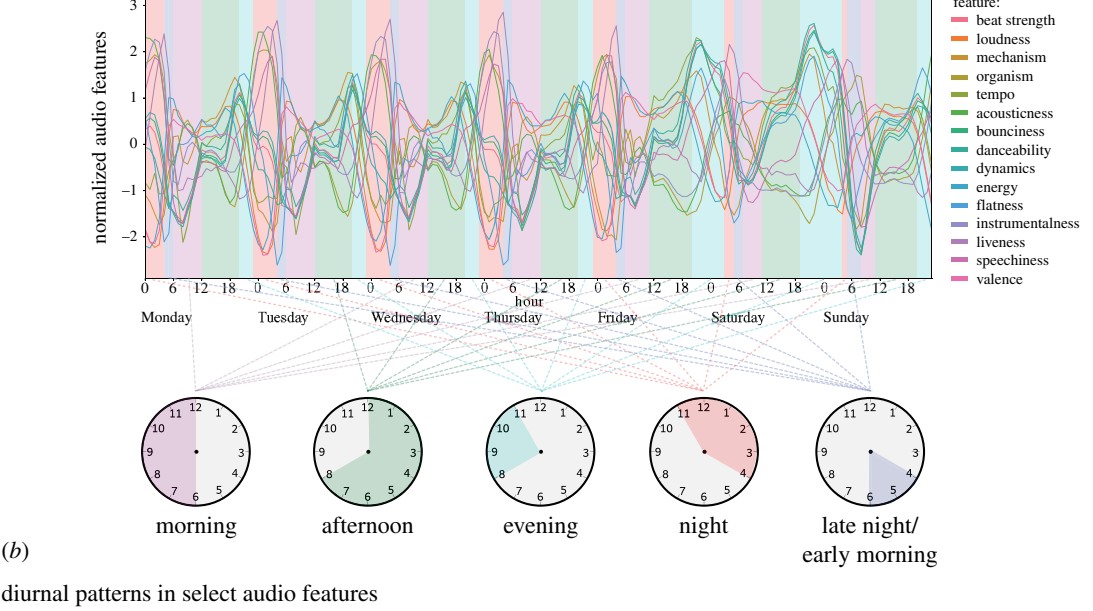

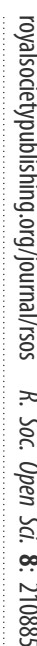

(b)

diurnal patterns in select audio features

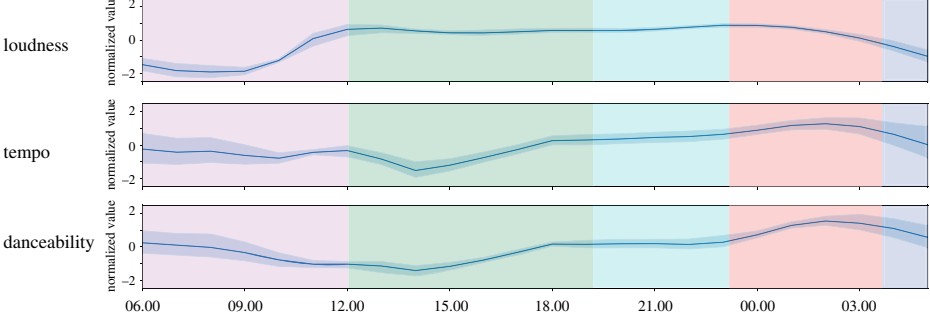

**Figure 1.** Diurnal patterns in audio features map to five subdivisions of the day. (a) Normalized audio features plotted across the entire week. Using k-means clustering we found an optimal division into five distinct clusters, here shown as coloured overlays on the plot. We labelled the clusters' temporal occurrence by first calculating the mode onset, and then using a descriptive term of the time of day. The clusters' temporal occurrences were always sequential and covered highly similar subdivisions of the day. We illustrate the mode onset and offset of each subdivision on the clock illustrations. (b) To highlight the diurnal cycles in audio features we here show three selected audio features, tempo, loudness and danceability, across 24 h starting at 06.00. The normalized audio features are shown as the mean over all weekdays, and the shaded area indicates the 95% confidence interval.

This data-driven approach allowed us to identify whether variations in audio features were predominantly driven by the time of the day or by the time of the week (e.g. weekday versus weekend).

## 2.1. Diurnal patterns in audio features map to five subdivisions of the day

Using k-means clustering we identified an optimal division of the hourly averaged audio features into five distinct clusters (figure 1a). Remarkably, the cluster labels of the hourly audio features exhibited a consistently sequential relationship across the week. Their temporal occurrence corresponds to subdivisions of the 24-h day, with some variations in the clusters' onset hours and lengths (illustrated in figure 1). We labelled clusters based on their overlap with commonly recognized times of the day: morning (mode = 06.00 h, s.d. = 0.73), afternoon (mode = 12.00 h, s.d. = 0.50), Evening (mode = 20.00 h, s.d. = 0.35), Night (mode = 23.00 h, s.d. = 1.61) and Late Night/early morning (mode = 04.00 h, s.d. = 0.45). All the individual audio features exhibited clear diurnal cycles as illustrated for tempo, loudness and danceability in figure 1b (for the remaining audio features, see electronic supplementary material, figure S1). Here, we can see how loudness, a measure how of perceptually loud a track is, exhibits an increase towards the end of the morning, then remains steady throughout most of the day

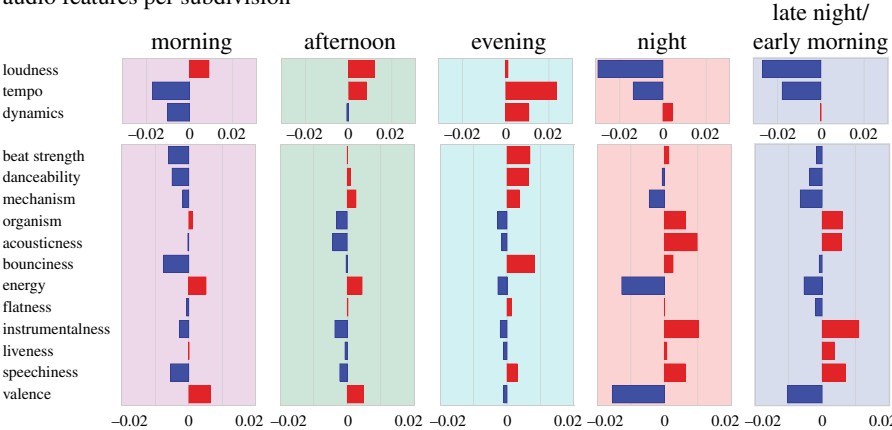

**Figure 2.** Audio feature values per subdivision. Here the cluster centroid values of the normalized audio features are shown, in relation to the grand average of the particular audio feature across the week. Red indicates a relative increase, and blue indicates a relative decrease. Note that loudness, tempo and dynamics have differing scales.

before tapering off towards the end of the night. Tempo and danceability exhibit a different pattern, with both having a mid-day lull at around 14.00 before peaking during the night.

In figure 2, we show how the subdivisions' audio features relate to the individual audio feature's grand mean. While these values are small in terms of musically perceivable differences (see §2.2) they serve as useful indicators of underlying trends. We found that the morning was characterized by an increase in loudness, valence and energy, yet with a decrease in tempo as compared to the grand mean of all audio features. In the afternoon, tempo increased markedly, with beat strength and danceability reaching average values. The highest tempo was found in the evening, together with peaks in beat strength and danceability. In the night both loudness and tempo fell to their lowest values. The late night/early morning exhibited a similar distribution, but with an increase in energy and valence as compared to the night.

Interestingly, the transition to the weekend did not impact the order of the subdivisions. During the weekends the evening subdivision increased in length, covering portions of the day that would normally be in the night subdivision (figure 1a). When considering the audio features for the evening, we saw a relative increase in tempo, beat strength, danceability and bounciness, pointing towards the evening containing tracks associated with dancing and partying. In response to the increase in length of the evening, the night subdivision had a later onset and shorter duration on Friday and Saturday night (03.00 and 04.00, respectively, compared to a weekday onset of 23.00). The late night/early morning subdivision also exhibited a slight shift in onset (05.00 on both Saturday and Sunday, compared to 04.00 on weekdays). The remaining subdivision exhibited highly similar onset points throughout the day.

Our results indicate that musical preferences, as defined by audio features, change cyclically and predictably throughout the day. As such, our study provides a conceptual replication of previous studies on music consumption finding general diurnal patterns using different, yet related, outcome measures [17,32].

## 2.2. Musically meaningful differences

While the subdivisions we identified were clearly quantitatively definable by the normalized audio feature values, it should be noted that the absolute audio feature values only exhibit perceptually small changes. For instance, the mean tempo lies in the range of 122.3–122.8 BPM for all subdivisions. Musically speaking, those are non-discernible differences. This indicates that people, in general, listen to a wide variety of music, as evident by the small changes in mean audio feature values between subdivisions and uniformly large standard deviations (for an overview see electronic supplementary material, table S1 and figure S2). Yet, on a population level, the difference in audio feature distributions changes enough for our data-driven approach to identify distinct subdivisions. This raises the question of whether the audio feature difference between the subdivisions is musically meaningful.

While the changes in mean audio feature values are perceptually small, the standard deviations are uniformly large (for a detailed overview, see electronic supplementary material, table S1). This indicates the continual presence of a wide distribution of audio feature values, and the musically meaningful difference between the subdivisions likely lies in the range of the distributions and in the unique combinations of audio feature characteristics of each subdivision. A possible interpretation here is that a song with audio features falling close to the mean values for a given subdivision is also likely to be found in another subdivision. However, a song exhibiting audio features close to the extreme of the distribution in a given subdivision is less likely to be found in another subdivision. In other words, we conjecture that whereas Every Breath You Take by The Police (a mid-tempo soft rock ballad) may be listened to uniformly throughout the day, Svefn-g-englar by Sigur Rós (a slow, ambient and dreamy song) may trend during the night.

Given that the musical difference between subdivisions lies in a shifting range of audio feature distributions, it is of interest to identify what causes, and contributes to, the shift in the distributions. One possible explanation could be that our data contain unique sub-populations listening to music at different times of the day. Due to our need for sleep, one is unlikely to find individuals continuously listening to music over a 24-h period, and the data analysed in this study is the result of a great number of individuals listening to music distributed over the entire day. As the MSSD does not contain demographic information, we were unable to address the impact of individual differences on musical preference. However, previous research on a similar dataset has shown an impact on baseline musical intensity preference by multiple demographic and individual attributes, such as age, sex, geographical region and chronotype (as defined by the 6-h interval wherein the individual was most active) [17]. Nonetheless, while the baseline changed between groups, the diurnal pattern of preferred musical intensity remained robust. For the four different chronotypes, only the night owls (individuals mostly active between 00.00 h and 06.00 h) presented with a significant difference in their diurnal pattern. The night owls exhibited a delayed increase of musical intensity in the morning as compared to the three other chronotypes. Hence, while individual differences likely contribute to baseline differences in diurnal musical preference, it is unlikely that they are the main explanatory factors. If this was the case, there should be a general consensus that some types of music are better suited to particular times of the day.

## 2.3. Awareness of diurnal musical preference in playlists

To examine if people consciously prefer certain types of music at particular times of the day, we designed an online study assessing general awareness of diurnal musical preference. We first trained an artificial neural network to classify the audio features of tracks to the five subdivisions found in the MSSD (for details see Methods). The fully trained network, performing at 97.16% accuracy on holdout data, was integrated with Spotify's API in an online app. Participants were then invited to get their playlists (a user-made collection of tracks) classified by the neural network, followed by a question as to whether they agreed with the network's prediction, or if they preferred to listen to their playlist at another subdivision of the day.

The participants ($N = 89$) tested the network on 253 playlists and agreed with the network's predicted subdivision for 63% of the playlists. To measure agreement, we calculated a circular error score based on distance from predicted subdivision to the participant's preferred subdivision (ranging from 0 to 2, for details see Methods). Overall, the mean error score was 0.55. The highest agreement was found for the evening (82.35%, error score = 0.21), followed by the night (69.77%, error score = 0.4), the morning (56.72%, error score = 0.7), the afternoon (44.74%, error score = 0.84) and the late night/early morning (35%, error score = 1.2). For all subdivisions the predicted classification had the highest level of agreement, indicating that the participants were more likely to agree with the prediction of the classifier than with a random prediction (see figure 3, panel 1). These results suggest that people are consciously aware, opinionated and in agreement with the predictions about which type of playlists they prefer for the different subdivisions of the day.

Interestingly, the least amount of agreement was found for the late night/early morning. In terms of general human behaviour this stretch of time around the break of dawn is the period where the least amount of people are awake [33,34]. Hence, it may be that the late night/early morning is a stretch of time wherein awake human activity is highly diverse. One would likely find three overarching groups of the population being awake and listening to music in the small hours of the night.

First, and most intuitively, the habitual listeners at these hours may be people working night shifts. Secondly, the late night/early morning is a transitory phase wherein night owls are just about to go to

(*a*)  awareness of diurnal musical preference

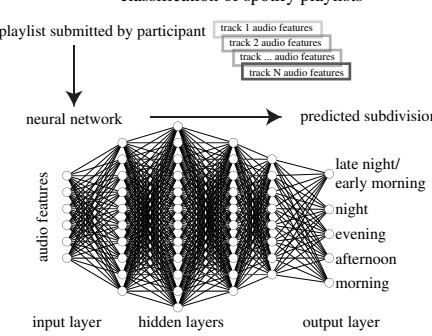

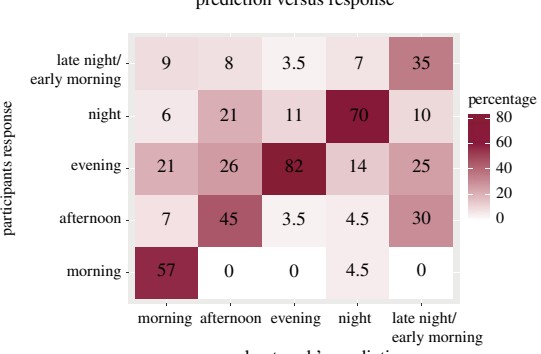

(*b*)  musical variability correlates with activity diversity

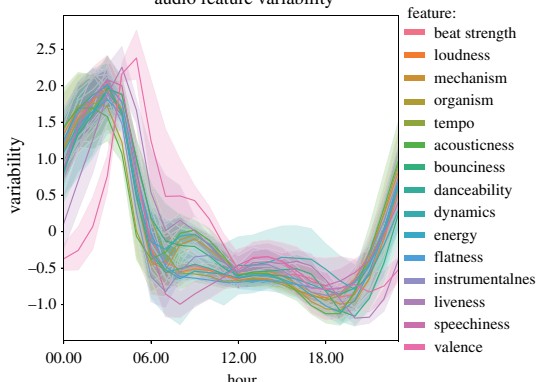

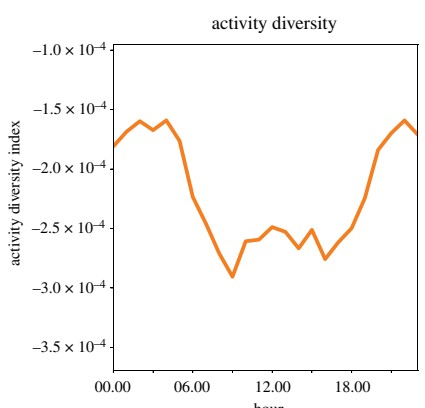

**Figure 3.** Classification of the daily listening habits and their correlation with daily activity diversity. (*a*) In the left panel we illustrate how we used a neural network to classify Spotify playlists to different musical subdivisions of the day. In the right panel, we show the agreement between the neural network's predictions and participants' responses. (*b*) In the left panel, we show how audio feature variability exhibits a strong peak in the late night/early morning. Each line corresponds to an audio feature, which is averaged per hour of the day over all days of the week. The shaded area indicates the 95% confidence interval. In the right panel, we show a moving average of the activity diversity over 24 h.

sleep, and morning larks are just getting up. Thirdly, one could imagine a group of the population being awake and listening to music during these hours as the result of an isolated event, perhaps returning from a party, pulling an all-nighter or needing to catch an early flight. Together, this raises the possibility that the late night/early morning is a stretch of time wherein a small population use music for more diverse purposes than the other subdivisions.

To assess the diversity of music being listened to, we calculated and normalized the standard deviations of all audio features. When collating these values over the 24 h of the day, this measure gives an indication of the relative diversity of music being listened to. As shown in figure 3, a clear diurnal pattern emerges, wherein the values remain steady throughout the morning and afternoon before rising sharply throughout the night with a peak and decrease during the late night/early morning. This observation indicates that musical preferences diverge during the night, with a broader range of audio features and hence types of music being listened to. This finding may in part explain the low level of agreement we found for predicted late night/early morning playlists.

To further explore the relation between diversity of music and human activity during the late night/ early morning, we used the ATUS to estimate activity diversity throughout the day [35]. This publicly available survey includes detailed data on time spent on various activities throughout the day. We used data from 2003 to 2019 and filtered out sleeping activities and missing data. This left us with 210 525 unique individuals, each with one full day of activities recorded. We estimated diversity of activity by calculating an index reflecting the proportional distribution of the top 50 activities for each hour of the day (for details, see Methods).

We found that the activity diversity index did indeed peak during the night; however, the peak was located a few hours before the mode onset of the late night/early morning subdivision, as shown in

**Table 1.** Results of survey on diurnal preference in representative tracks. Mean preference is a relative value indicating whether participants preferred to listen to e.g. morning tracks in the morning compared to the other four subdivisions of the day. Positive values indicate a preference for listening to the tracks during the model-predicted subdivision of the day, whereas negative values indicate a preference for listening to the tracks during a different subdivision of the day. Statistical significance was assessed with *t*-tests against zero and Bonferroni-corrected *p*-values.

|  | mean preference | s.d. | $t(175)$ | $p$ | 95% CI |
|---|---|---|---|---|---|
| morning | 4.34 | 19.93 | 2.89 | 0.022 | [1.37, 7.30] |
| afternoon | −4.38 | 24.84 | −2.34 | 0.102 | [−8.08, −0.69] |
| evening | 0.34 | 21.63 | 0.21 | 0.999 | [−2.87, 3.56] |
| night | −4.64 | 19.09 | −3.23 | 0.008 | [−7.48, −1.80] |
| late night/early morning | 10.92 | 25.14 | 5.76 | <0.001 | [7.18, 14.66] |

figure 3. Nonetheless, we found a significant moderate correlation between the activity diversity and the mean audio feature variability (Pearson's $R = 0.47$, $p = 0.022$). As the MSSD does not contain demographic data, we are unable to ascertain whether this difference could be ascribed to regional or cultural factors. In addition, due to the *post hoc* nature of this analysis, we did not investigate the relative impact of individual audio features on the correlation. However, the correlation between diversity of activity and diversity of musical preference shows that combining multiple sources of population-level big data is a promising avenue for uncovering diurnal patterns of behaviour.

## 2.4. No clear diurnal musical preference for individual tracks

Having established the existence of diurnal fluctuations in musical preference both in the general listening habits contained within the MSSD and in user-created playlists, we wanted to see if these preferences exist also in individual tracks. To that end, we designed an online survey where we used a combination of data-driven and qualitative approaches to select three representative tracks from each of the five subdivisions (for details, see Methods). Participants ($N = 176$) were asked to listen to an excerpt of each track and rate their likelihood of listening to the tracks at each subdivision of the day. The participants significantly preferred morning and late night/early morning tracks at their intended subdivision ($p = 0.022$, and $p < 0.001$, respectively, Bonferroni-corrected), yet for the night a negative preference was found ($p = 0.008$). For the afternoon and evening no significant preference was found (table 1).

While these results may at first glance seem counterintuitive, they in fact reflect and highlight our finding that diurnal musical preference is driven by changes in the range of audio features associated with the subdivisions of the day. While some musical tracks may present with a clear association to a subdivision, tracks laying close to the mean audio feature values are suitable at any time of the day. In addition, it is important to highlight that music listening is rarely a one-track event. This finding is in line with more general research on music consumption, indicating that people listen to music as a session-like activity, whereas listening to a single track is mainly associated with special occurrences [36,37]. Hence, playlists of multiple tracks, but not individual tracks, reflect diurnal musical preference.

Nonetheless, these results highlight the feasibility of finding, and making, music that is uniquely suited to a particular time of the day. In the Sangita Makaranda, an ancient treatise on Hindustani classical music, this is highlighted in the lines 'one who sings knowing the proper time remains happy. By singing ragas at the wrong time one ill-treats them. Listening to them, one become impoverished and sees the length of one's life reduced' [38]. Hence, the highly significant diurnal preference found for our late night/early morning tracks could represent finding exactly some of the music highly suited for this particular time of the day.

Interestingly, there is likely market pressure against music composed for a particular time of the day. The music industry is increasingly geared towards a pay-per-stream model, wherein each separate stream of a track generates monetary reward. Tracks that are more likely to be preferred at any time of the day should have a higher probability of getting streamed and hence generate more revenue. This may produce market pressure towards homogenization of music [39]. Analysis of large datasets of audio features such as the MSSD could provide valuable insight into this process.

# 3. Discussion

In this work, we have shown that the rhythms of daily life are accompanied by fluctuations in musical preference. We show that the diurnal patterns of audio features in music can be treated as five distinct subdivisions of the day, with the musically meaningful distinction between them found in the range and distribution of the musical audio features. Our follow-up studies indicate that individuals hold a general awareness and agreement of diurnal musical preference in playlists consisting of multiple tracks, but that single tracks do not necessarily elicit the same diurnal associations. Taken together, this points to the circadian rhythms governing life being reflected in the highly individualized and often subjective preference for music.

The next step in this line of research would be to examine the degree to which the diurnal patterns documented herein reflect universal psychological phenomena in music perception. As previously discussed, some types of music often occur at a specific time of the day and often with a clear link to activities, with perhaps lullabies being a prime example. As lullabies are intended to ease falling asleep, they tend to occur at night and have been found to have partly universal features such as reduced tempo [40–42]. If similar time-dependent songs could be collected into a database, it would then be highly interesting to investigate if the audio features of such songs match up with the features that drive the time-of-day preferences uncovered herein. Here, the Spotify API's ability to search user-made playlists for name and description is a highly productive approach, as shown in a recent study uncovering a large amount of variation in sleep music [43].

While the diurnal patterns in musical audio features uncovered in this work are robust and consistent with previous research, there are nonetheless limitations to highlight. In particular, our analysis has not addressed demographical and geographical influence on the results. In part, this is due to the lack of both demographical and individual-level information in the MSSD, and due to our data being based on Spotify, biasing the findings towards the population with access to the service. This means that our results are inherently biased towards Western culture, and we are unable to investigate factors such as age and occupation which have previously been found to impact listening behaviour [44,45]. We would encourage future research to work on combining datasets from multiple providers, such as QQ Music, Gaana and Boomplay, to ensure a wider geographical and cultural representation. Collating such datasets would require collaboration with the music streaming industry and work on harmonizing the many approaches to calculating musically meaningful audio features [46,47]. In addition, the audio features may miss out on nuances in high-level understanding of musical behaviour such as the behavioural functions of the music, and aspects of emotional content [48,49].

As a final note, we would highlight that this project has been carried out using open-source software and publicly available data, with all analysis and programming performed on laptop computers, and that the data collection processes in this work were undertaken without incurring any direct costs. This shows how the availability of digital traces from online activity can be used to investigate human behaviour by scientists both affiliated and independent alike [50].

# 4. Material and methods

Our pre-processing and analysis were performed in Python and R. Code and data used in this project are available at github.com/OleAd/DiurnalSpotify.

## 4.1. The music streaming sessions dataset

To identify diurnal patterns of audio features we used The MSSD, which is publicly available from CrowdAI (https://www.crowdai.org/organizers/spotify/challenges/spotify-sequential-skip-prediction-challenge). For a detailed description see Brost *et al.* [26]. The dataset contains approximately 150 million streaming sessions of up to 20 tracks per session, with a total of approximately 3.7 million unique tracks being listened to. From the MSSD we kept streaming events (i.e. playback of a track) wherein the listener listened to the most of the track, and with a maximum of one seek-forward or -backward operation. This ensured that we only analysed musical tracks that were actually listened to. We excluded streaming events with missing or corrupt information of date and hour of the day, leading to 2 026 529 205 included streaming events in our dataset. We extracted a subset of the track metadata available in the MSSD describing low-level and high-level audio features of the included tracks (for an overview see Audio features).

To analyse how audio features change as a function of weekday and hour of the day, we calculated mean and standard deviation for audio features binned per unique hour of the week. This calculation was performed using Dask in Python v. 3.7.6 [51]. To assess any recurring patterns, we used $k$-means clustering to partition the data into $k$ clusters with minimized within-cluster variance. With the $k$-means algorithm selecting the optimal $k$ depends on the intended outcome in terms of data interpretation. Here, we intended to see if visually observed cyclic variation in auditory features would be best described by a weekday division (e.g. business day versus weekend), or within-day division (e.g. day versus night).

We selected the optimal $k$ by first clustering normalized audio features data over a range of values for $k$ (2–24) while calculating the inertia (within-cluster sum-of-squares) for each cluster. This calculation was performed using the KMeans function in DaskML, with an oversampling factor of 2 and 100 maximum iterations per $k$. We used the elbow method, an observational method wherein the inertia values are visually inspected and the point of the curve wherein the decrease in inertia for each increase in $k$ notably flattens, to identify the optimal partition of our dataset at $k = 5$, and re-calculated the clustering at 1000 maximum iterations. Data normalization was performed using the StandardScaler function in scikit-learn. This function normalizes data by removing the mean and scaling to unit variance, such that the standard score $z$ of sample $x$ is calculated as $z = (x - u)/s$, wherein $u$ is the mean and $s$ is the standard deviation. Following selection $k = 5$, we used an inverse transformation on the normalized cluster centroids to get the mean and standard deviation of audio features in each cluster.

Following the clustering we inspected the cluster label of each unique hour of the week and found that they exhibited a sequential relationship across the 168 h of the week, indicating a repeating cycle of clusters consistently tracking the 24-h day. To find the hours of the day corresponding to the clusters, we calculated the modal hour of the day for the onset of each occurrence of a cluster. In order to compare the audio features between clusters, we subtracted the mean values of all audio features from each cluster, thus giving us a relative number indicating if audio feature values in one cluster were above or below the average. Due to the large sample size, no inferential statistics were used, as even a miniscule difference in mean values between clusters would be deemed statistically significant. Instead, we qualitatively assessed the relative difference between averaged audio features in each cluster.

## 4.2. Audio features

Currently, the music information retrieval field relies on multiple approaches to calculating audio features, such as for instance the MIR toolbox. The audio features available from the Spotify API are based on calculations originally developed by the Echo Nest, wherein complex audio features, such as danceability, are derived from non-public algorithms [52]. As such, we are unable to quantify exactly which musical, perceptual and acoustical parameters contribute to a particular audio feature. To enable comparisons between different streaming services we would encourage the development of shared and open-sourced audio feature calculations.

For an overview of the audio features used in the project, in table 2 we provide a short overview based on Spotify's API reference [53]. Notably, in the original paper describing the MSSD, and in the accompanying description of the dataset, descriptions of the audio features are not included [26]. Instead, they refer to Spotify's audio feature description as part of their API reference manual, wherein some of the features are not documented. To gain an overview of the non-documented audio features we went through multiple blogs, documented code projects and the Echo Nest Analyzer documentation from prior to Spotify's acquisition of Echo Nest.

## 4.3. Probing diurnal preference in playlists and single tracks

To identify the extent to which people are aware of diurnal musical preference we designed two follow-up studies. One, wherein we probed playlists, and one wherein we probed single tracks. As the MSSD does not contain detailed information on the tracks included, such as track and artist name nor a link to a preview.mp3-file, we decided to build two datasets of Spotify tracks and their associated audio features. The first, and smallest, dataset was used for selecting single tracks, whereas the second, larger dataset was used for training the artificial neural network.

## 4.4. Building datasets of Spotify tracks

The first dataset consisted of popular tracks, collected using the Spotify API through Spotipy [54]. We selected top playlists at different timepoints and weekdays during week 17 of 2020, both in the global category and for

**Table 2.** Overview of audio features. In this table we summarize the audio features present in the MSSD and available from the Spotify API.

| audio feature | description |
| --- | --- |
| energy | A perceptual measure of intensity, covering perceptual features including dynamic range, loudness, timbre, onset rate and general entropy. Ranges from 0 to 1, with higher values indicating a higher energy. |
| danceability | A measure of how suitable a track is for dancing. Includes musical features such as tempo, rhythmic stability, beat strength and regularity. Ranges from 0 to 1, with higher values indicating higher danceability. |
| acousticness | A measure of how likely a track is acoustic, meaning music that is solely or primarily performed using non-electric or non-electronic instruments. |
| dynamic range | This measure is not specified. It is a positive value and, in all likelihood, reflects the dynamic range in a track, given in decibels (dB). We assume this value measures the mean dynamic range in the track, as a measure of the range in amplitude within a given window of samples. |
| beat strength | This measure is undocumented. It ranges between 0 and 1. The audio feature likely describes the rhythmic and perceptual characteristics contributing to the perceived clarity of the beat in a track. |
| liveness | A measure of how likely a track is performed live. No information given on the perceptual features involved in this audio feature is given. |
| organism | This measure is undocumented. It ranges between 0 and 1. This audio feature is likely a compound measure of multiple perceptual features and other audio features, indicating if a track is perceived as 'organic', in the sense that tempo and acousticness may exhibit a higher variability, dynamic range may be high. |
| mechanism | This measure is undocumented. It ranges between 0 and 1. This audio feature is likely a compound measure of multiple perceptual features and other features, indicating if a track is perceived as 'mechanic', as opposed to the previously mentioned organism. |
| valence | A measure describing musical positiveness, ranging from 0 to 1. A higher valence indicates a track that may be described as 'happy, cheerful, euphoric', a lower value indicates a track that may be described as 'sad, depressed, angry'. No information is given on the calculation of this audio feature. |
| bounciness | This measure is undocumented. It ranges between 0 and 1. This audio feature is likely a compound measure based on perceptual and acoustical factors contributing to a sense of 'bounce' in the track, likely involving calculating attack slopes and general length of musical onsets. |
| flatness | This measure is undocumented. It ranges between 0 and 1. This audio feature is likely a compound measure based on perceptual and acoustical factors contributing to a sense of 'flatness' in the track, likely involving a low density of events and little auditory transients. |
| loudness | A measure of the overall loudness of a track. Spotify specifies this in dB without an explicit reference. We, therefore, assume that it is given in dB full scale (dBFS) wherein 0 indicates clipping level. The loudness is stated to range between −60 and 0, but positive values are seen in the MSSD. |
| speechiness | A measure of the presence of spoken words in a track. The value ranges from 0 to 1. Spotify specifies that values above 0.66 indicate a high likelihood of a track consisting entirely of spoken words. No information is given on the perceptual features involved in calculating this audio feature. |

(*Continued.*)

**Table 2.** (*Continued.*)

| audio feature | description |
|---|---|
| instrumentalness | A measure of the likelihood of a track containing no vocals. The value ranges from 0 to 1. No information is given on the perceptual features involved in calculating this audio feature. |
| tempo | A measure of the speed, or pace, of a track, given in beats per minute (BPM). This value is calculated based on the average beat duration in a track. |

specific countries (US, GB, DK, NO, DE, SE, FI, CA, MX, NI, PT). From these playlists we extracted tracks, removed duplicates, and extracted the available audio features using Spotify's API (danceability, energy, loudness, speechiness, acousticness, instrumentalness, liveness, valence, tempo). Some of the tracks did not contain music but were instead field recordings of weather (such as rain and thunder) or spoken words only. These tracks were subsequently removed from the dataset, giving a total of 25 178 unique tracks. This dataset is available on https://github.com/OleAd/DiurnalSpotify/tree/master/datasets.

The second dataset was created by augmenting the first dataset with multiple online datasets of Spotify audio features found on repositories such as Kaggle and Data.world. These extra datasets include, among others, the global most-streamed tracks of 2019, and the Billboard weekly hot 100 singles from 1958–2019. Together, this resulted in a dataset containing 473 610 unique tracks. The dataset is available on https://github.com/OleAd/DiurnalSpotify/tree/master/datasets.

## 4.5. Classifying playlists with an artificial neural network

To classify playlists, we trained an artificial neural network to classify single tracks to one of the five subdivisions of the day and implemented the trained network in an online app which predicted the predominant subdivision of participants' Spotify playlists.

To select tracks for a training dataset we first calculated descriptive statistics for the audio features of all tracks included in the five subdivisions identified in the MSSD. From these we created a threshold, wherein for danceability, energy, liveness and valence we first selected the mean value for a given subdivision. We then selected tracks from the second dataset wherein the audio features were above or below this threshold dependent on a given subdivision's audio features' relation to the general mean. For example, the mean valence value for the afternoon subdivision was higher than the general mean valence. Then, only tracks with valence values above this threshold were selected for the training dataset. Speechiness, acousticness and instrumentalness exhibited highly non-normal distributions and were not taken into consideration when selecting tracks for the training dataset.

This resulted in a selection of data containing the following numbers of tracks: late night/early morning—15 020 tracks, morning—10 058 tracks, afternoon—11 350 tracks, evening—1860 tracks, night—3197 tracks. From this selection, we split the data into a training dataset and a holdout dataset with an 80/20-split (available on https://github.com/OleAd/DiurnalSpotify/tree/master/data). Since the dataset was unbalanced between the subdivisions, we calculated class weights to be used in training the neural network. Due to the intended goal of using the neural network to classify Spotify playlist using audio features from the Spotify API in real-time, we did not standardize the audio features. Instead, we used a remapping function wherein we linearly transformed audio features to the range (−1,1). For loudness, we set less than or equal to −60 to map to −1 and greater than or equal to 12 to map to 1, and for tempo we set less than or equal to 40 to map to −1 and greater than or equal to 220 to map to 1.

We constructed the neural network using the Keras high-level API in Tensorflow v. 2.0 [55]. The network takes six audio features (danceability, energy, loudness, liveness, valence and tempo, due to the highly non-normal distributions of speechiness, acousticness and instrumentalness) as inputs into four fully connected hidden layers of 64, 128, 64 and 32 nodes, respectively, with an output softmax layer of five nodes corresponding to the five subdivisions of the data. All activations in the hidden layers were set to sigmoid (a necessity due to our aim of using the trained network in an online app, wherein Tensorflow.js at the time of implementation exhibited issues with other common activation functions such as relu). We split the training dataset into a training and validation dataset with another 80/20-split and trained the network using the Adam optimizer with a categorical cross-entropy loss function and an initial learning rate of 0.0001. To avoid overfitting, we used a 30% dropout between the hidden layers, as well as an L1-regularizer of 0.01 on the last three hidden layers. We trained the network in batch sizes of 32 and shuffled the data between each epoch. Training was monitored using an early stopping rule halting

training once validation accuracy had not improved over 30 epochs. The trained neural network performed at 97.16% accuracy with 0.0738 loss on holdout data.

To use the trained neural network in an online app, we converted it to run in Tensorflow.js (TFJS) [56]. This is a JavaScript variant of Tensorflow. We built a responsive app using Node.js, express, TFJS and handlebars, and hosted the app on the Google App Engine, with data written to a MongoDB database running on the MongoDB Atlas service. Participants would enter the app, read instructions and then be redirected to user authorization using Spotify's API, which would give us permission to read the participant's playlists. The participants then selected a playlist, which were analysed client-side using our trained neural network. The network classified each track in the playlist separately and calculated a mean probability of each subdivision. The participants were presented with the label of the subdivision with the highest probability and asked whether they agreed with the classification. A short text with information as well as a graph of probabilities was also presented. If the participants did not agree with the classification, they were given an opportunity to select their own classification. Following this, the participant's response, the playlist and its tracks IDs and the neural network's classification were written to the database. The participants were then given the opportunity to answer a voluntary question on their emotional state when listening to the playlist, and/or to try another playlist. Data were collected over four weeks, with 89 participants contributing with 253 playlists. No demographical or geographical information was collected. Participants were geographically limited to regions where Spotify is available. Notably, this excludes multiple regions in Africa and parts of Asia. The code for the online app is available at github.com/OleAd/SpotifyPlaylistClassifierApp.

The responses were compared to the network's classification by calculating a circular error distance. Here, the error value is the shortest step-wise distance between the participant's response and the network's classification as given in the following equation: $CircErr = argmin(res - pred, \ 5 + pred - res)$ where res is the participant's response and pred is the network's classification both in the range [1,5]. Given that there were five possible classification results, the maximum error value is 2. This value was calculated for the entire sample and for each predicted subdivision individual.

## 4.6. Diurnal preference in single tracks

To select representative tracks for each subdivision we here used the first, smaller, dataset consisting of, recently, featured tracks. We first used a similar thresholding process as with the neural network training dataset, with the addition of adding or subtracting 10% of the standard deviation of the audio features' threshold value, dependent on the subdivision's mean audio feature values' relation to the general mean. For example, the mean valence value for the afternoon subdivision was higher than the general valence mean value. Then, a threshold is calculated as the mean valence value for all tracks belonging to this subdivision plus 10% of the standard deviation of valence for this subdivision, and only tracks with valence values above this threshold were chosen as a possible representative track. This approach ensured that we selected candidate tracks from the extremes of the distributions of audio feature values for each subdivision. We ended up with the following numbers of candidate tracks: early morning/late night—418 tracks, morning—293 tracks, afternoon—553 tracks, evening—77 tracks, night—64 tracks. As some of these tracks did not have an associated 30-s preview URL, some tracks were discarded at this stage.

To reduce the number of tracks we performed PCA on standardized audio features and selected the five tracks with the highest value of the first component, five tracks around the middle value and five tracks with the lowest value. From these 15 tracks, two of the authors chose four tracks each, for which we acquired a full-length high-quality audio file. For these tracks, we plotted the first two components from the PCA to visualize a clear separability between the tracks which qualitatively guided the selection of four tracks per subdivision. To reduce the four tracks to the three we needed for the survey, we recruited 11 co-workers at the Center for Functionally Integrative Neuroscience and the Center for Music in the Brain at Aarhus University and asked them to rate how familiar they were with excerpts from each of the 20 tracks. We then removed the most popular track for each subdivision, leaving us with three tracks per subdivisions, i.e. 15 tracks in total.

A total of 248 participants completed the survey. Fifty-four participants provided no rating and 18 participants rated less than 66% of the tracks. These participants were excluded resulting in a final sample size of $N = 176$ (mean age 28.5 years, s.d. = 7.6). We presented the 15 tracks (three per subdivision of the day) in randomized order. Participants rated for every track how likely they would be to listen to this type of music during the five different subdivisions of the day on a continuous slider ranging from 1 to 101.

For statistical analyses, each track's rating was compared to the ratings of the tracks in the other subdivisions of the day. For example, the mean rating of all afternoon, evening, night and late

night/early morning tracks per participant was subtracted from the three individual morning tracks. All participants' means of these difference measures of each subdivision were then compared against zero with values larger than zero indicating a preference for listening to the tracks during the model-predicted subdivision of the day, and values less than zero indicating a preference for listening to the tracks during a different subdivision of the day. Statistical significance was assessed with $t$-tests against zero. $p$-values were Bonferroni-corrected for five comparisons.

## 4.7. Correlating activity diversity and musical variability

To examine a link between the diversity of activities and musical variability, we used data from the ATUS for 2003–2019. In the ATUS, activities are reported with minute-level accuracy. To harmonize the data with the hour-level accuracy in the MSSD we rounded reported activities to full hours and used a forward-fill method to expand the data to cover each hour of the 24-hour day. The data held activities reported from 210 586 unique reporters. We filtered out activities related to sleeping (activity ID 010 101, 010 199 and 010 102), and missing data (activity ID 500 101 and higher).

To assess the diversity of activities we first calculated the proportion of each unique activity for each individual hour of the day. We then selected the top 50 activities in descending order and calculated the descending differences in proportion. As an example, if the 1st activity has an occurrence proportion of 0.4, and the 2nd activity has an occurrence proportion of 0.3, the difference is −0.1. This leads to a list of 49 difference values, for which we calculate the median value. The resulting value lies in the range 0 to −1, with a higher value indicating a smaller difference between the proportions, and vice versa. We take this value as the activity diversity index, which we correlated with musical variability calculated as the per-hour mean of the standard deviation of all normalized audio features.

Ethics. At Center for Music in the Brain, Department of Clinical Medicine, Aarhus University, Denmark, ethical approval for non-medical research is governed by the CFIN IRB. In Denmark, research that does not collect nor store personally identifiable or sensitive information are exempt from IRB approval, which we confirmed in correspondence with the IRB (e-mail, 30 April 2020). Our data collection was performed in compliance with the Danish Code of Conduct for Research Integrity and Aarhus University's Policy for research integrity, freedom of research and responsible conduct of research. All data collected was anonymized and included no personally identifiable information. For participants submitting playlists for classification, we used Spotify's OAuth v. 2.0 authorization flow to request permission for viewing public and private playlists only, thereby ensuring that no personally identifiable information such as playlist names were stored in our system. The authorization permission was valid for 60 minutes only, whereafter it was deleted.

Data accessibility. The Music Streaming Sessions Dataset is available from: https://www.crowdai.org/organizers/spotify/challenges/spotify-sequential-skip-prediction-challenge. The American Time Use Survey dataset for 2003–2019 is available from: https://www.bls.gov/tus/datafiles-0319.html. Generalized Python scripts used for analysing both datasets are available on https://www.github.com/OleAd/DiurnalSpotify. The data collected and generated as part of this project are found in the same repository, and have been archived within the Zenodo repository: https://doi.org/10.5281/zenodo.5562678 [57] . Python and R scripts for creating the plots shown in this paper, as well as the trained neural network, are found in the same repository. Code for the Node.js application used for classification of playlists is available at https://github.com/OleAd/SpotifyPlaylistClassifierApp and have been archived within the Zenodo repository: https://doi.org/10.5281/zenodo.5562691 [58].

Authors' contributions. O.A.H. developed the study concept, pre-processed and analysed the MSSD dataset. O.A.H. and J.S. designed the questionnaire and playlist study under supervision from P.V. J.S. analysed data from the questionnaire study. O.A.H. developed, coded and implemented the classification of playlists study. All authors contributed to data interpretation. O.A.H. drafted the manuscript and figures, and J.S. and P.V. provided critical revisions. All authors approved the final version of the manuscript for submission.

Competing interests. The authors report no conflict of interest.

Funding. O.A.H., J.S. and P.V. are supported as part of the Center for Music in the Brain, Danish National Research Foundation (grant no. DNRF117). J.S. is supported by an Erwin Schrödinger Fellowship from the Austrian Science Fund (grant no. FWF J4288-B27).

Acknowledgements. The authors wish to thank co-workers and students at the Center for Music in the Brain and the Center of Functionally Integrative Neuroscience, and the anonymous referees for their useful suggestions.

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
