## [Peer Review File · Royal Society Open Science]

Review History

RSOS-210885.R0 (Original submission)

Review form: Reviewer 1

Is the manuscript scientifically sound in its present form?

Yes

Are the interpretations and conclusions justified by the results?

Yes

Is the language acceptable?

Yes

Do you have any ethical concerns with this paper?

No

Have you any concerns about statistical analyses in this paper?

No

Recommendation?

Accept with minor revision (please list in comments)

Comments to the Author(s)

Review of "Diurnal fluctuations in musical preference"

This paper presents an interesting analysis of a publicly available dataset of Spotify usage. Listening behavior was classified into 5 categories which are reliably associated with times of day throughout the week, and with distributions of musical features. In an experiment, users submitted their own playlists for time-of-day classification by a neural net, which generally was able to predict the users' time-of-day classification, based on the musical features of the songs in the playlists. Last, participants in another experiment rated the fit of specific songs to each time-of-day classification; surprisingly, these ratings were not terribly predictive of songs' *actual* time-of-day classifications, suggesting that diurnal patterns in musical preferences are driven by groups of songs (playlists, albums, etc) rather than specific songs.

I found the paper to be well-written, methodologically appropriate, and the interpretations and presentation to be rigorous. The topic is interesting and appropriate for RSOS.

Here are some comments and suggestions for a revision:

In Figure 1 and the initial discussion of this result in the first section of the Results, it should be made more explicit sooner in the manuscript that the effect sizes that are visualized and discussed, in terms of means of the musical features, are absolutely tiny. This does not undermine the results, but it's important to help the reader understand what results are actually being presented.

It also raises the possibility that there might be better approaches than simply looking at the mean values. One alternative is that the extremes of the distribution at a given time point might be more informative; this could be added to the paper and analyzed in tandem with the central tendency version of the analyses. It might also be informative to visualize the distribution of values over the different time points. Is it always a neat normal distribution, or might there be other peaks that could be modelled with some kind of mixture model or similar? Or could one code the absence/presence of a particular song in different times of day rather than taking the mean of a noisily coded feature across a large number of streams?

The results in section 2.3 are particularly compelling in validating aspects of the previous analysis with human listeners, especially in light of the very small effect sizes in the main analysis. I would suggest devoting more space to this aspect of the results in the Discussion and set it up in the Introduction, potentially with reference to other studies of listeners' high-level understanding of aspects of musical behavior that are not actually present in the acoustic forms of music, such as the emotional content of music (e.g., Sievers et al., 2013, PNAS) or the behavioral functions of the music (e.g., Mehr et al., 2018, Current Biology).

One other general question is the degree to which the phenomena documented in this large database reflect universal psychological phenomena in music perception and cognition. One simple thing to measure would be to look at general types of songs that turn up at a given time of day (like lullabies, which are for sleeping, which usually happens at night) and seeing if the features that characterize those songs universally (like slower tempos and smoother melodic contours; Mehr et al., 2019, Science; Unyk et al., 1992, Psych of Music) match up to those that characterize time-of-day classifications in the Spotify data reported here. This could be done for a variety of different types of songs; if the features that universally characterize a particular type of song are the same features that drive time-of-day preferences for music, it would provide unique evidence that naturalistic user data in music streaming reflects general psychological properties of musical interest and response.

Minor comments:

p8, line 52 "...clusters exhibited a consistently sequential relationship across the week"
If I understood correctly, the k-means clustering was just on the audio features and not any temporal features? It would be good to state this, if so.

p.10, line 47

Could you elaborate on what exactly was conceptually replicated here? I.e., is it just that there are diurnal patterns generally, or that the specific features shift in similar ways?

p. 11, line 35

The examples of specific songs are nice, however, it would be good to include (perhaps in brackets) a brief elaboration of what is normal or extreme about the audio features in these songs.

In all, this was an interesting paper and thanks to the authors for a fun read.

Review form: Reviewer 2

Is the manuscript scientifically sound in its present form?

No

Are the interpretations and conclusions justified by the results?

No

Is the language acceptable?

Yes

Do you have any ethical concerns with this paper?

No

Have you any concerns about statistical analyses in this paper?

No

Recommendation?

Major revision is needed (please make suggestions in comments)

Comments to the Author(s)

The manuscript presents an interesting study on the variance of music consumption behavior as reflected in Spotify listening histories over each hour of the week. To this end, the authors leverage the MSSD. While most results are not overly surprising, the work contributes to research on data analysis carried out in the fields of music information retrieval and music recommender systems.

The paper is generally well-written, language-wise. However, several parts are quite confusing. Some of this confusion certainly originates from the fact that the results are presented before the actual methodology is introduced (something I am frankly not used to, even though I already reviewed hundreds of scientific papers).

Nevertheless, the authors should do a better job in clarifying important methodological details earlier in the manuscript. Also, they should provide already in the introduction a very clear statement of their actual contributions and individual experiments carried out. For instance, when reading the manuscript, it was very unclear to me in which experiments which kind of data

(or which datasets) were used, in particular, when and why some experiments used individual tracks and some used "playlists". I first thought the authors confused "playlists" with "listening sessions", which are very different things; because MSSD contains only listening sessions, not hand-crafted playlists. Only later it became somewhat clear that the authors additionally gathered users' playlists and used them for additional experiments/user studies. All in all, it is hard to understand which experiments use MSSD, which rely on Spotify users' individual user-generated content, and which rely on data from other sources. All this confusion could have been avoided by clearly mentioning in the introduction explicit research questions and main methodological aspects (e.g., statistical data analysis for the large-scale data-driven study, user study involving analysis of Spotify users' own playlists, etc.).

One part of the manuscript particularly suited to exemplify this problem is Section 2.4 "Playlists, but not individual tracks, reflect diurnal musical preference". From the text provided in this section, it does not even become clear how and why *playlists* were analyzed. In fact, the text in this whole section does not even mention the word "playlist" again. Therefore, from the running text in Section 2.4, I do not see any empirical evidence for the author's statement made in the section name. Table 1 is no more informative either. E.g., "Mean" numbers are reported, but what the scale is (on which values are these means computed?) remains unclear.

I very much appreciate the authors' discussion of limitations, in particular, the inherent data "bias towards western culture" and the inability "to investigate factors such as age". However, both could be resolved to some extent by repeating the statistical analysis on additional datasets such as LFM-1b or MLHD (for details, see my comments below in "Additional remarks and suggestions").

Regarding the study on playlists, I am wondering why the authors did not consider the standardized Million Playlist Dataset (also by Spotify), see:
<https://www.aicrowd.com/challenges/spotify-million-playlist-dataset-challenge>.

Related to the features under investigation, the authors rely on Spotify's audio features even though they are intransparent and their computation is a black box (which authors honestly admit). I, therefore, think that authors should also consider and investigate semantically more meaningful features instead of barely graspable concepts of "organism" or "bounciness". What comes to mind quickly is genre information (which is provided by Spotify too).

Additional remarks and suggestions:

- The authors claim that MSSD is the "to-date largest publicly available dataset on music listening behaviour including information on when, during the day, a particular track is listened to". I am not sure that this statement is true. Please also consider mentioning the MLHD ([https://ddmal.music.mcgill.ca/research/The_Music_Listening_Histories_Dataset_\(MLHD\)/](https://ddmal.music.mcgill.ca/research/The_Music_Listening_Histories_Dataset_(MLHD)/)) and the LFM-1b and LFM-2b datasets (<http://www.cp.jku.at/datasets/LFM-1b/> and <http://www.cp.jku.at/datasets/LFM-2b/>).
- A screenshot of the UI used in the user studies would be highly appreciated to support the textual explanations.
- Details about the survey participants should be given: how were they recruited? which platform was used (Amazon MTurk)? are there demographic biases?
- Please include in each figure (and/or caption) a very explicit indication about which values are shown on the axes (in particular the y-axes). For instance, Figure 1A just says "Audio features" on the y-axis with a numeric range between -3 and 3, which is unclear (because most Spotify features have a range between 0 and 1). The same issue can be seen for Figure 1B, and others. Also include, in Figure 1A, a legend telling the reader which color represents which audio feature.

- It would be very interesting to include a visualization similar to Figure 2 but showing working days vs. weekends.
- "This indicates that people, in general, listen to a wide variety of music, as evident by the small changes in mean audio feature values between subdivisions.": Why is a small change in the audio features' *mean* evidence for a wide variety? A *large standard deviation* would rather provide such evidence, in my opinion.
- "absolute audio feature values only exhibit qualitatively small changes. For instance, the mean Tempo lies in the range of 122.3 to 122.8 BPM": I guess "qualitatively" should read "quantitatively" here.
- "to ensure a wider geographical and cultural representation. Collating such datasets would require collaboration with the music streaming industry...": There do exist datasets that cover users with a wide variety of cultural backgrounds/ different countries. I suggest having a look at the LFM-1b, LFM-2b, and MLHD, for instance (see above).
- Please add in Section 4.1 a link to the MSSD.
- Please clarify the undefined terms and unclear choices you used/made throughout the manuscript (e.g., in Sec 2.1, what is an "optimal division" (optimal in which sense? according to which measure/metric? how do you guarantee optimality?); "playlist" versus "(listening) session" should be clearly defined and made clear in which experiment each is used (see my comment above); Section 4.5: "the network takes six audio features" => why only 6 and why those 6?
- In Sec 4.7, the difference between proportions of adjacent activities in the ranking is expressed in positive numbers first ("0.1"), but then it's said: "The resulting value lies in the range 0 to -1." Please be consistent.
- Correct a few grammar mistakes: "audio features relates to" => "relate to"; "Red indicate a relative" => "indicates"; "analyzed music tracks that was" => "were"
- From the (rather vague) description of the approach in Sec 4.1 (e.g., using terms such as "the data" without saying exactly which data/features), I do not get why the "five clusters identified by the k-means clustering exhibited a sequential relationship across the 168 hours". Was the sequential characteristic considered when computing k-means? Was one clustering for each of the 168-hour-bins computed and clusters tracked over time? If so, how were they tracked? If not, how can you obtain a sequential relationship from the single clustering of all the data?
- Instead of saying that the StandardScaler function of scikit-learn was used, rather provide a formula of what it does.
- Talking about formulas, I would highly appreciate a formula instead of (or in addition to) the lengthy description of "circular error distance" at the end of Sec 4.5.

Decision letter (RSOS-210885.R0)

Dear Dr Heggli

The Editors assigned to your paper RSOS-210885 "Diurnal fluctuations in musical preference" have now received comments from reviewers and would like you to revise the paper in accordance with the reviewer comments and any comments from the Editors. Please note this decision does not guarantee eventual acceptance.

We invite you to respond to the comments supplied below and revise your manuscript. Below the referees' and Editors' comments (where applicable) we provide additional requirements.

Final acceptance of your manuscript is dependent on these requirements being met. We provide guidance below to help you prepare your revision.

Please submit your revised manuscript and required files (see below) no later than 21 days from today's (ie 03-Aug-2021) date. Note: the ScholarOne system will 'lock' if submission of the revision is attempted 21 or more days after the deadline. If you do not think you will be able to meet this deadline please contact the editorial office immediately.

on behalf of Professor Joydeep Bhattacharya (Associate Editor) and Essi Viding (Subject Editor)
openscience@royalsociety.org

Reviewer comments to Author:
Reviewer: 1
Comments to the Author(s)
Review of "Diurnal fluctuations in musical preference"

This paper presents an interesting analysis of a publicly available dataset of Spotify usage. Listening behavior was classified into 5 categories which are reliably associated with times of day throughout the week, and with distributions of musical features. In an experiment, users submitted their own playlists for time-of-day classification by a neural net, which generally was able to predict the users' time-of-day classification, based on the musical features of the songs in the playlists. Last, participants in another experiment rated the fit of specific songs to each time-of-day classification; surprisingly, these ratings were not terribly predictive of songs' *actual* time-of-day classifications, suggesting that diurnal patterns in musical preferences are driven by groups of songs (playlists, albums, etc) rather than specific songs.

I found the paper to be well-written, methodologically appropriate, and the interpretations and presentation to be rigorous. The topic is interesting and appropriate for RSOS.

Here are some comments and suggestions for a revision:

In Figure 1 and the initial discussion of this result in the first section of the Results, it should be made more explicit sooner in the manuscript that the effect sizes that are visualised and

discussed, in terms of means of the musical features, are absolutely tiny. This does not undermine the results, but it's important to help the reader understand what results are actually being presented.

It also raises the possibility that there might be better approaches than simply looking at the mean values. One alternative is that the extremes of the distribution at a given time point might be more informative; this could be added to the paper and analyzed in tandem with the central tendency version of the analyses. It might also be informative to visualize the distribution of values over the different time point. Is it always a neat normal distribution, or might there be other peaks that could be modelled with some kind of mixture model or similar? Or could one code the absence/presence of a particular song in different times of day rather than taking the mean of a noisily coded feature across a large number of streams?

The results in section 2.3 are particularly compelling in validating aspects of the previous analysis with human listeners, especially in light of the very small effect sizes in the main analysis. I would suggest devoting more space to this aspect of the results in the Discussion and set it up in the Introduction, potentially with reference to other studies of listeners' high-level understanding of aspects of musical behavior that are not actually present in the acoustic forms of music, such as the emotional content of music (e.g., Sievers et al., 2013, PNAS) or the behavioral functions of the music (e.g., Mehr et al., 2018, Current Biology).

One other general question is the degree to which the phenomena documented in this large database reflect universal psychological phenomena in music perception and cognition. One simple thing to measure would be to look at general types of songs that turn up at a given time of day (like lullabies, which are for sleeping, which usually happens at night) and seeing if the features that characterize those songs universally (like slower tempos and smoother melodic contours; Mehr et al., 2019, Science; Unyk et al., 1992, Psych of Music) match up to those that characterize time-of-day classifications in the Spotify data reported here. This could be done for a variety of different types of songs; if the features that universally characterize a particular type of song are the same features that drive time-of-day preferences for music, it would provide unique evidence that naturalistic user data in music streaming reflects general psychological properties of musical interest and response.

Minor comments:

p8, line 52 "...clusters exhibited a consistently sequential relationship across the week"

If I understood correctly, the k-means clustering was just on the audio features and not any temporal features? It would be good to state this, if so.

p.10, line 47

Could you elaborate on what exactly was conceptually replicated here? I.e., is it just that there are diurnal patterns generally, or that the specific features shift in similar ways?

p. 11, line 35

The examples of specific songs are nice, however, it would be good to include (perhaps in brackets) a brief elaboration of what is normal or extreme about the audio features in these songs.

In all, this was an interesting paper and thanks to the authors for a fun read.

Reviewer: 2

Comments to the Author(s)

The manuscript presents an interesting study on the variance of music consumption behavior as reflected in Spotify listening histories over each hour of the week. To this end, the authors leverage the MSSD. While most results are not overly surprising, the work contributes to research on data analysis carried out in the fields of music information retrieval and music recommender systems.

The paper is generally well-written, language-wise. However, several parts are quite confusing. Some of this confusion certainly originates from the fact that the results are presented before the actual methodology is introduced (something I am frankly not used to, even though I already reviewed hundreds of scientific papers).

Nevertheless, the authors should do a better job in clarifying important methodological details earlier in the manuscript. Also, they should provide already in the introduction a very clear statement of their actual contributions and individual experiments carried out. For instance, when reading the manuscript, it was very unclear to me in which experiments which kind of data (or which datasets) were used, in particular, when and why some experiments used individual tracks and some used "playlists". I first thought the authors confused "playlists" with "listening sessions", which are very different things; because MSSD contains only listening sessions, not hand-crafted playlists. Only later it became somewhat clear that the authors additionally gathered users' playlists and used them for additional experiments/user studies. All in all, it is hard to understand which experiments use MSSD, which rely on Spotify users' individual user-generated content, and which rely on data from other sources. All this confusion could have been avoided by clearly mentioning in the introduction explicit research questions and main methodological aspects (e.g., statistical data analysis for the large-scale data-driven study, user study involving analysis of Spotify users' own playlists, etc.).

One part of the manuscript particularly suited to exemplify this problem is Section 2.4 "Playlists, but not individual tracks, reflect diurnal musical preference". From the text provided in this section, it does not even become clear how and why *playlists* were analyzed. In fact, the text in this whole section does not even mention the word "playlist" again. Therefore, from the running text in Section 2.4, I do not see any empirical evidence for the author's statement made in the section name. Table 1 is no more informative either. E.g., "Mean" numbers are reported, but what the scale is (on which values are these means computed?) remains unclear.

I very much appreciate the authors' discussion of limitations, in particular, the inherent data "bias towards western culture" and the inability "to investigate factors such as age". However, both could be resolved to some extent by repeating the statistical analysis on additional datasets such as LFM-1b or MLHD (for details, see my comments below in "Additional remarks and suggestions").

Regarding the study on playlists, I am wondering why the authors did not consider the standardized Million Playlist Dataset (also by Spotify), see:

<https://www.aicrowd.com/challenges/spotify-million-playlist-dataset-challenge>.

Related to the features under investigation, the authors rely on Spotify's audio features even though they are intransparent and their computation is a black box (which authors honestly admit). I, therefore, think that authors should also consider and investigate semantically more meaningful features instead of barely graspable concepts of "organism" or "bounciness". What comes to mind quickly is genre information (which is provided by Spotify too).

Additional remarks and suggestions:

- The authors claim that MSSD is the "to-date largest publicly available dataset on music listening behaviour including information on when, during the day, a particular track is listened to". I am not sure that this statement is true. Please also consider mentioning the MLHD ([https://ddmal.music.mcgill.ca/research/The_Music_Listening_Histories_Dataset_\(MLHD\)/](https://ddmal.music.mcgill.ca/research/The_Music_Listening_Histories_Dataset_(MLHD)/)) and the LFM-1b and LFM-2b datasets (<http://www.cp.jku.at/datasets/LFM-1b/> and <http://www.cp.jku.at/datasets/LFM-2b/>).

- A screenshot of the UI used in the user studies would be highly appreciated to support the textual explanations.

- Details about the survey participants should be given: how were they recruited? which platform was used (Amazon MTurk)? are there demographic biases?

- Please include in each figure (and/or caption) a very explicit indication about which values are shown on the axes (in particular the y-axes). For instance, Figure 1A just says "Audio features" on the y-axis with a numeric range between -3 and 3, which is unclear (because most Spotify features have a range between 0 and 1). The same issue can be seen for Figure 1B, and others.

Also include, in Figure 1A, a legend telling the reader which color represents which audio feature.

- It would be very interesting to include a visualization similar to Figure 2 but showing working days vs. weekends.

- "This indicates that people, in general, listen to a wide variety of music, as evident by the small changes in mean audio feature values between subdivisions.": Why is a small change in the audio features' *mean* evidence for a wide variety? A *large standard deviation* would rather provide such evidence, in my opinion.

- "absolute audio feature values only exhibit qualitatively small changes. For instance, the mean Tempo lies in the range of 122.3 to 122.8 BPM": I guess "qualitatively" should read "quantitatively" here.

- "to ensure a wider geographical and cultural representation. Collating such datasets would require collaboration with the music streaming industry...": There do exist datasets that cover users with a wide variety of cultural backgrounds/different countries. I suggest having a look at the LFM-1b, LFM-2b, and MLHD, for instance (see above).

- Please add in Section 4.1 a link to the MSSD.

- Please clarify the undefined terms and unclear choices you used/made throughout the manuscript (e.g., in Sec 2.1, what is an "optimal division" (optimal in which sense? according to which measure/metric? how do you guarantee optimality?); "playlist" versus "(listening) session" should be clearly defined and made clear in which experiment each is used (see my comment above); Section 4.5: "the network takes six audio features" => why only 6 and why those 6?

- In Sec 4.7, the difference between proportions of adjacent activities in the ranking is expressed in positive numbers first ("0.1"), but then it's said: "The resulting value lies in the range 0 to -1." Please be consistent.

- Correct a few grammar mistakes: "audio features relates to" => "relate to"; "Red indicate a relative" => "indicates"; "analyzed music tracks that was" => "were"

- From the (rather vague) description of the approach in Sec 4.1 (e.g., using terms such as "the data" without saying exactly which data/features), I do not get why the "five clusters identified by the k-means clustering exhibited a sequential relationship across the 168 hours". Was the sequential characteristic considered when computing k-means? Was one clustering for each of the 168-hour-bins computed and clusters tracked over time? If so, how were they tracked? If not, how can you obtain a sequential relationship from the single clustering of all the data?

- Instead of saying that the StandardScaler function of scikit-learn was used, rather provide a formula of what it does.

- Talking about formulas, I would highly appreciate a formula instead of (or in addition to) the lengthy description of "circular error distance" at the end of Sec 4.5.

===PREPARING YOUR MANUSCRIPT===

===PREPARING YOUR REVISION IN SCHOLARONE===

-- Ensure that your data access statement meets the requirements at <https://royalsociety.org/journals/authors/author-guidelines/#data>. You should ensure that you cite the dataset in your reference list. If you have deposited data etc in the Dryad repository, please include both the 'For publication' link and 'For review' link at this stage.

-- If you have uploaded ESM files, please ensure you follow the guidance at <https://royalsociety.org/journals/authors/author-guidelines/#supplementary-material> to

include a suitable title and informative caption. An example of appropriate titling and captioning may be found at https://figshare.com/articles/Table_S2_from_Is_there_a_trade-off_between_peak_performance_and_performance_breadth_across_temperatures_for_aerobic_sc_ope_in_teleost_fishes_/3843624.

Author's Response to Decision Letter for (RSOS-210885.R0)

See Appendix A.

RSOS-210885.R1 (Revision)

Review form: Reviewer 2

Is the manuscript scientifically sound in its present form?

Yes

Are the interpretations and conclusions justified by the results?

Yes

Is the language acceptable?

Yes

Do you have any ethical concerns with this paper?

No

Have you any concerns about statistical analyses in this paper?

No

Recommendation?

Accept with minor revision (please list in comments)

Comments to the Author(s)

The revised manuscript has improved much over the original version. Authors have addressed my comments in their response, mostly in a satisfactory manner, and updated their manuscript accordingly.

I also have to clarify that, given my background in computer and data science, I reviewed the manuscript of course taking such a perspective, acknowledging that expectations and requirements in the medical and psychological disciplines might be different (for instance, most researchers in CS/DS/AI would complain about the use of the very simple k-means clustering algorithm when nowadays ample alternatives exist). From this point of view, I found a bit strange what the authors say in their response to justify why they did not use other datasets to investigate whether results generalize. While it is true that LFM-* and MLHD do not come with audio features, matching the tracks to Spotify URIs and fetching the respective audio features

from the corresponding Spotify API endpoints certainly does not take years. A Master's student of mine managed to do this in a couple of days.

Notwithstanding, this is (still) a very interesting paper with some novel findings!

On the other hand, there are still a few issues that need to be addressed before acceptance:

- I assume there will be a final copyediting step. When referencing papers that appeared in conference proceedings, authors currently use an incomplete format, in particular, they mention "editors", but don't give the names of those editors (just the author names). They also don't give the corresponding pages of the paper in the proceedings. Furthermore, some conferences are abbreviated, while others are not: e.g., "27. Vigiensoni G, Fujinaga I, editors. The Music Listening Histories Dataset. ISMIR; 2017." vs. "28. Schedl M, editor The lfm-1b dataset for music retrieval and recommendation. Proceedings of the 2016 ACM on international conference on multimedia retrieval; 2016.". All of this is in stark contrast to the complete and seemingly correct format authors use for journal articles.

- The manuscript should be checked for consistency in US vs. UK spelling.

- On page 20/63, authors now mention the LFM-2b dataset, but do not include the corresponding reference:

Alessandro B. Melchiorre, Navid Rekabsaz, Emilia Parada-Cabaleiro, Stefan Brandl, Oleg Lesota, Markus Schedl: Investigating gender fairness of recommendation algorithms in the music domain. *Inf. Process. Manag.* 58(5): 102666 (2021)

- On page 30/63, authors should also report the mean accuracy achieved by their NN over all classes (they do so only for each class separately, and only the mean error is provided overall).

- On the same page, when discussing the results of the NN experiments, I suggest to be a bit more cautious. For instance, I don't think it's fair to say that results indicate that "the participants in general agreed with predicted subdivisions". From the results it seems they rather "were more likely to agree with the prediction of the classifier than with a random prediction" (a dumb baseline classifier that randomly picks a class).

- Again here, "These results suggest that people are ... to a high degree in agreement about which type of playlists they prefer". This sentence can be confusing as it might imply that people are in agreement with each other, not with the predictions. I suggest to add "in agreement with the predictions".

- On page 33/63, "...both in general listening habits and in user-created playlists, we wanted to see if these preferences exist also in individual tracks". When reading "general listening habits" I first thought that this means computed on individual tracks. But that's what authors set up to investigate next. I suggest to be more precise and indicate (again) how these "general listening habits" are defined/computed.

- Page 36/63: "... patterns ... reflect*s*"

- Page 42/63: Authors should add a reference to the MIR toolbox, at least to the webpage: <https://www.jyu.fi/hytk/fi/laitokset/mutku/en/research/materials/mirtoolbox/> (I think there is a paper reference, too).

- On page 47/63, authors say that the network contains five fully-connected hidden layers.

However, authors report just four values for the number of units in each layer. Looking at Figure 3, indeed, it seems that are just 4 hidden layers + 1 input + 1 output layer.

- On page 48/63, the UI used in the user study is described in pretty much detail. I still think a screenshot should be given here to better understand the look&feel of the UI. At the very least, authors should explicitly refer to the additional material and add a link to screenshots of the UI.

Review form: Reviewer 3

Is the manuscript scientifically sound in its present form?

Yes

Are the interpretations and conclusions justified by the results?

Yes

Is the language acceptable?

Yes

Do you have any ethical concerns with this paper?

No

Have you any concerns about statistical analyses in this paper?

No

Recommendation?

Accept with minor revision (please list in comments)

Comments to the Author(s)

The authors have addressed the key issues we raised in our initial review, and we both think it is mostly ready for publication.

Below are a few additional points the authors may wish to incorporate (all page numbers are to the pages of the proof document, e.g., out of 63 pages):

- p18 line23 - The sentence "In a broader perspective..." seems a bit tacked on, and feels like it either needs more elaboration, or to be removed.
- You can probably better integrate your new text on page 20, line48-51 and the following section, since you say that the MSSD contains pre-calculated musical audio features two sentences in a row.
- On another read of the whole paper, one thing that stood out was that the jump from the introduction to the results section felt like it needed more preparation. I.e., you introduced the fact that diurnal patterns occur in nature and human behaviour, and music specifically, and then say you are going to use the MSSD to investigate the latter, and briefly describe this dataset... and then you jump straight into the analysis. It felt like it needed 2-3 more sentences to reiterate what exactly it is that we do not know and stand to gain from the study. E.g., the final couple sentences were basically "Prior research... has shown..." and it felt like it needed something like "But, it is not clear whether..." and a "We now use the MSSD, along with X computational methods to provide some answers to these questions".
- The y-axis text in Fig.1b is small and hard to read
- page 24, line 18 - Here you add a sentence in light of our previous comment. This helps, but actually I would recommend both being slightly more direct AND would bring this up at the start of the paragraph. E.g., from the start of the paragraph "In Figure 2 we show... . While these mean differences are themselves too small to be perceived (see section 2.2), here, they serve as indicators of underlying trends. We found...".
- page 13, line 48 - You conjecture that a song with characteristics close to mean would be common across times of day, whereas songs with characteristics closer to the extremes in feature-space would be more likely unique to single distributions... this seems intuitive, but perhaps not trivially true. I assume you did not run any analyses on this? This section could benefit from even a simple analysis to at least partly backup this claim empirically, or at least clarify that this is a speculation and not a finding from your analyses.

Again thanks to the authors, and all the best for the paper.

Decision letter (RSOS-210885.R1)

Dear Dr Heggli

On behalf of the Editors, we are pleased to inform you that your Manuscript RSOS-210885.R1 "Diurnal fluctuations in musical preference" has been accepted for publication in Royal Society Open Science subject to minor revision in accordance with the referees' reports. Please find the referees' comments along with any feedback from the Editors below my signature.

Please submit your revised manuscript and required files (see below) no later than 7 days from today's (ie 06-Oct-2021) date. Note: the ScholarOne system will 'lock' if submission of the revision is attempted 7 or more days after the deadline. If you do not think you will be able to meet this deadline please contact the editorial office immediately.

on behalf of Professor Joydeep Bhattacharya (Associate Editor) and Essi Viding (Subject Editor)
openscience@royalsociety.org

Reviewer comments to Author:

Reviewer: 3

Comments to the Author(s)

The authors have addressed the key issues we raised in our initial review, and we both think it is mostly ready for publication.

Below are a few additional points the authors may wish to incorporate (all page numbers are to the pages of the proof document, e.g., out of 63 pages):

- p18 line23 - The sentence "In a broader perspective..." seems a bit tacked on, and feels like it either needs more elaboration, or to be removed.
- You can probably better integrate your new text on page 20, line48-51 and the following section, since you say that the MSSD contains pre-calculated musical audio features two sentences in a row.
- On another read of the whole paper, one thing that stood out was that the jump from the introduction to the results section felt like it needed more preparation. I.e., you introduced the fact that diurnal patterns occur in nature and human behaviour, and music specifically, and then say you are going to use the MSSD to investigate the latter, and briefly describe this dataset... and then you jump straight into the analysis. It felt like it needed 2-3 more sentences to reiterate what exactly it is that we do not know and stand to gain from the study. E.g., the final couple sentences were basically "Prior research... has shown..." and it felt like it needed something like "But, it is not clear whether..." and a "We now use the MSSD, along with X computational methods to provide some answers to these questions".
- The y-axis text in Fig.1b is small and hard to read
- page 24, line 18 - Here you add a sentence in light of our previous comment. This helps, but actually I would recommend both being slightly more direct AND would bring this up at the start of the paragraph. E.g., from the start of the paragraph "In Figure 2 we show... . While these mean differences are themselves too small to be perceived (see section 2.2), here, they serve as indicators of underlying trends. We found...".
- page 13, line 48 - You conjecture that a song with characteristics close to mean would be common across times of day, whereas songs with characteristics closer to the extremes in feature-space would be more likely unique to single distributions... this seems intuitive, but perhaps not trivially true. I assume you did not run any analyses on this? This section could benefit from even a simple analysis to at least partly backup this claim empirically, or at least clarify that this is a speculation and not a finding from your analyses.

Again thanks to the authors, and all the best for the paper.

Reviewer: 2

Comments to the Author(s)

The revised manuscript has improved much over the original version. Authors have addressed my comments in their response, mostly in a satisfactory manner, and updated their manuscript accordingly.

I also have to clarify that, given my background in computer and data science, I reviewed the manuscript of course taking such a perspective, acknowledging that expectations and requirements in the medical and psychological disciplines might be different (for instance, most researchers in CS/DS/AI would complain about the use of the very simple k-means clustering algorithm when nowadays ample alternatives exist). From this point of view, I found a bit strange what the authors say in their response to justify why they did not use other datasets to investigate whether results generalize. While it is true that LFM-* and MLHD do not come with audio features, matching the tracks to Spotify URIs and fetching the respective audio features from the corresponding Spotify API endpoints certainly does not take years. A Master's student of mine managed to do this in a couple of days.

Notwithstanding, this is (still) a very interesting paper with some novel findings!

On the other hand, there are still a few issues that need to be addressed before acceptance:

- I assume there will be a final copyediting step. When referencing papers that appeared in conference proceedings, authors currently use an incomplete format, in particular, they mention "editors", but don't give the names of those editors (just the author names). They also don't give the corresponding pages of the paper in the proceedings. Furthermore, some conferences are abbreviated, while others are not: e.g., "27. Vigiensoni G, Fujinaga I, editors. The Music Listening Histories Dataset. ISMIR; 2017." vs. "28. Schedl M, editor The lfm-1b dataset for music retrieval and recommendation. Proceedings of the 2016 ACM on international conference on multimedia retrieval; 2016.". All of this is in stark contrast to the complete and seemingly correct format authors use for journal articles.

- The manuscript should be checked for consistency in US vs. UK spelling.

- On page 20/63, authors now mention the LFM-2b dataset, but do not include the corresponding reference:

Alessandro B. Melchiorre, Navid Rekabsaz, Emilia Parada-Cabaleiro, Stefan Brandl, Oleg Lesota, Markus Schedl: Investigating gender fairness of recommendation algorithms in the music domain. *Inf. Process. Manag.* 58(5): 102666 (2021)

- On page 30/63, authors should also report the mean accuracy achieved by their NN over all classes (they do so only for each class separately, and only the mean error is provided overall).

- On the same page, when discussing the results of the NN experiments, I suggest to be a bit more cautious. For instance, I don't think it's fair to say that results indicate that "the participants in general agreed with predicted subdivisions". From the results it seems they rather "were more likely to agree with the prediction of the classifier than with a random prediction" (a dumb baseline classifier that randomly picks a class).

- Again here, "These results suggest that people are ... to a high degree in agreement about which type of playlists they prefer". This sentence can be confusing as it might imply that people are in agreement with each other, not with the predictions. I suggest to add "in agreement with the predictions".

- On page 33/63, "...both in general listening habits and in user-created playlists, we wanted to see if these preferences exist also in individual tracks". When reading "general listening habits" I first thought that this means computed on individual tracks. But that's what authors set up to investigate next. I suggest to be more precise and indicate (again) how these "general listening habits" are defined/computed.

- Page 36/63: "... patterns ... reflect*s*"

- Page 42/63: Authors should add a reference to the MIR toolbox, at least to the webpage: <https://www.jyu.fi/hytk/fi/laitokset/mutku/en/research/materials/mirtoolbox/> (I think there is a paper reference, too).

- On page 47/63, authors say that the network contains five fully-connected hidden layers. However, authors report just four values for the number of units in each layer. Looking at Figure 3, indeed, it seems that are just 4 hidden layers + 1 input + 1 output layer.

- On page 48/63, the UI used in the user study is described in pretty much detail. I still think a screenshot should be given here to better understand the look&feel of the UI. At the very least, authors should explicitly refer to the additional material and add a link to screenshots of the UI.

===PREPARING YOUR MANUSCRIPT===

===PREPARING YOUR REVISION IN SCHOLARONE===

- If you are providing image files for potential cover images, please upload these at this step, and inform the editorial office you have done so. You must hold the copyright to any image provided.
- A copy of your point-by-point response to referees and Editors. This will expedite the preparation of your proof.

- Ensure that your data access statement meets the requirements at <https://royalsociety.org/journals/authors/author-guidelines/#data>. You should ensure that you cite the dataset in your reference list. If you have deposited data etc in the Dryad repository, please only include the 'For publication' link at this stage. You should remove the 'For review' link.
- If you are requesting an article processing charge waiver, you must select the relevant waiver option (if requesting a discretionary waiver, the form should have been uploaded at Step 3 'File upload' above).
- If you have uploaded ESM files, please ensure you follow the guidance at <https://royalsociety.org/journals/authors/author-guidelines/#supplementary-material> to include a suitable title and informative caption. An example of appropriate titling and captioning may be found at https://figshare.com/articles/Table_S2_from_Is_there_a_trade-off_between_peak_performance_and_performance_breadth_across_temperatures_for_aerobic_scope_in_teleost_fishes_/3843624.

Author's Response to Decision Letter for (RSOS-210885.R1)

See Appendix B.

Decision letter (RSOS-210885.R2)

Dear Dr Heggli,

I am pleased to inform you that your manuscript entitled "Diurnal fluctuations in musical preference" is now accepted for publication in Royal Society Open Science.

on behalf of Professor Joydeep Bhattacharya (Associate Editor) and Essi Viding (Subject Editor)
openscience@royalsociety.org

Appendix A

Dear Professor Joydeep Bhattacharya and Professor Essi Viding,

We have now worked through the reviewers' highly relevant and constructive comments. We have improved the clarity of the manuscript by better differentiating between our three studies, as well as between analyses and interpretations that relate to single tracks, listening sessions, and playlists. Additionally, we added relevant details to the explanation of the k -means procedure.

Following suggestions from Reviewer 1, we have added a supplementary figure consisting of 15 subplots showing individual fluctuations in audio feature distribution widths. We have also updated all other figures and included additional labels for better clarity.

We are thankful to the reviewers' comments relating to alternative analyses and datasets (e.g., LFM-1b or MLHD as mentioned by Reviewer 2). We have discussed these datasets at appropriate locations in the revised manuscript and added a new paragraph about future research perspectives in the discussion.

To address some of the specific comments from the reviewers we have provided a new script (SML_distributions.py) consisting of 300 lines of code, which is now available at GitHub, as well as made the entire code base for our real-time classifier application available in a separate repository (<https://github.com/OleAd/SpotifyPlaylistClassifierApp>).

We believe these changes has strengthened the manuscript, and hope that our revision and detailed replies to both reviewers (see below) are satisfactory. We look forward to proceeding with our manuscript.

With kind regards on behalf of the authors,
Ole Adrian Heggli

Response to reviewer 1

This paper presents an interesting analysis of a publicly available dataset of Spotify usage. Listening behavior was classified into 5 categories which are reliably associated with times of day throughout the week, and with distributions of musical features. In an experiment, users submitted their own playlists for time-of-day classification by a neural net, which generally was able to predict the users' time-of-day classification, based on the musical features of the songs in the playlists. Last, participants in another experiment rated the fit of specific songs to each time-of-day classification; surprisingly, these ratings were not terribly predictive of songs' *actual* time-of-day classifications, suggesting that diurnal patterns in musical preferences are driven by groups of songs (playlists, albums, etc) rather than specific songs.

I found the paper to be well-written, methodologically appropriate, and the interpretations and presentation to be rigorous. The topic is interesting and appropriate for RSOS.

Thank you!

Here are some comments and suggestions for a revision:

In Figure 1 and the initial discussion of this result in the first section of the Results, it should be made more explicit sooner in the manuscript that the effect sizes that are visualised and discussed, in terms of means of the musical features, are absolutely tiny. This does not undermine the results, but it's important to help the reader understand what results are actually being presented.

We agree, and have added the following sentence in the paragraph presenting Figure 2: *"However, in terms of musically perceivable differences, these fluctuations were small."*

It also raises the possibility that there might be better approaches than simply looking at the mean values. One alternative is that the extremes of the distribution at a given time point might be more informative; this could be added to the paper and analyzed in tandem with the central tendency version of the analyses. It might also be informative to visualize the distribution of values over the different time point. Is it always a neat normal distribution, or might there be other peaks that could be modelled with some kind of mixture model or similar? Or could one code the absence/presence of a particular song in different times of day rather than taking the mean of a noisily coded feature across a large number of streams?

Thank you for these useful suggestions! We definitely agree that much of what is interesting in the diurnal fluctuations likely lies in the extremes of the distributions.

In investigating this we ran into a few technological unresolved challenges. One approach to looking into the extremes of the distributions would be to look at the kurtosis. However, in Dask the kurtosis calculation is only implemented for single series, and not for dataframes. This means that while we are able to parallelize and perform the necessary out-of-core calculations for a single series, the entire set would require an estimated 2-3 weeks of compute time which we at the moment are unable to provide.

However, we were able to calculate descriptive statistics such as mean, standard deviation, and percentiles. These calculations are not computationally costly, but gated by read/write-speed, and takes around 9 hours for each subdivision. To visualize the extremes of the

distributions we decided to use the distance between the 5th and 95th percentile as indications of the width of the distributions. We then plotted these individually for each audio feature, and some interesting patterns can be observed. As an investigation into the individual audio features is beyond the scope of the current manuscript, we decided to include this in a Supplementary Figure, with the following figure text:

“Supplementary Figure 2: Distribution width overview. In this figure, the width of the audio feature distributions per subdivision is shown, as the distance between the 5th and 95th percentile of each feature. In general, we can observe that the distribution widths peak at different subdivision for each audio feature, with the majority peaking at Night and Late Night/Early Morning (such as Tempo, Speechiness, and Valence), with some other peaking in the Afternoon (such as Danceability and Bounciness).”

Similarly, plotting the full distributions at different time points would be very helpful in gaining an intuitive sense of the diurnal variation in audio features, yet we are unable to do so due to memory issues. Some of these challenges could be solved using commercially available software and distributed clusters. However, we have refrained from doing so, as one of the motivations for this project was to explore what can be done with big data on consumer-grade laptops, to show how students and independent researchers could approach research large-scale human behaviour.

We very much like the idea of coding the absence/presence of particular tracks during different times of the day. As a track may occur in all subdivision, we would think an outcome measure here would be a track's likelihood of occurrence for each given subdivision. Such an analysis would make very good sense in combination with other metadata such as the title of the track, the name of the artist, and genre, allowing us to answer questions such as what genre is most associated with the night, or identifying if there are tracks that only occur in specific subdivision. However, in the MSSD this type of metadata is not included, and an unreversible hash is applied to the track ID value, meaning we are unable to acquire this information.

The results in section 2.3 are particularly compelling in validating aspects of the previous analysis with human listeners, especially in light of the very small effect sizes in the main analysis. I would suggest devoting more space to this aspect of the results in the Discussion and set it up in the Introduction, potentially with reference to other studies of listeners' high-level understanding of aspects of musical behavior that are not actually present in the acoustic forms of music, such as the emotional content of music (e.g., Sievers et al., 2013, PNAS) or the behavioral functions of the music (e.g., Mehr et al., 2018, Current Biology).

Thank you for this suggestion! While we find this aspect to be highly interesting, we feel that our data lacks the necessary demographic information to fully comment on this aspect. However, we believe that music streaming data could be a fruitful avenue into addressing the non-acoustic embeddings of musical behaviour. In particular, in related work we often see people using descriptive language to name and describe their playlists, which could serve as a route of inferring the behavioural functions of their collections of music. We hope to address this in future work, and have added the following sentence to the discussion:

“In addition, the audio features may miss out nuances in high-level understanding of musical behaviour such as the behavioural functions of the music, and aspects of emotional content (Mehr, Singh, York, Glowacki, & Krasnow, 2018; Sievers, Polansky, Casey, & Wheatley, 2013).”

One other general question is the degree to which the phenomena documented in this large database reflect universal psychological phenomena in music perception and cognition. One simple thing to measure would be to look at general types of songs that turn up at a given time of day (like lullabies, which are for sleeping, which usually happens at night) and seeing if the features that characterize those songs universally (like slower tempos and smoother melodic contours; Mehr et al., 2019, Science; Unyk et al., 1992, Psych of Music) match up to those that characterize time-of-day classifications in the Spotify data reported here. This could be done for a variety of different types of songs; if the features that universally characterize a particular type of song are the same features that drive time-of-day preferences for music, it would provide unique evidence that naturalistic user data in music streaming reflects general psychological properties of musical interest and response.

This is an excellent suggestion, which we hope to address in future studies. Lullabies are perhaps the prime target here, due to their clear time-of-the-day association. An interesting question is whether it is the time-of-the-day or the linked activity's time association that drives it, a question that is hard to untangle. Here, user-generated playlists may serve as a good starting point for building datasets to explore this question. We have added a paragraph in the discussion pointing this out:

“A next step in this line of research would be to examine the degree to which the diurnal patterns documented herein reflects universal psychological phenomena in music perception. As previously discussed, some types of music often occur at a specific time of the day and often with a clear link to activities, with perhaps lullabies being a prime example. As lullabies are intended to ease falling asleep, they tend to occur at night, and have been found to have partly universal features such as reduced tempo (Bainbridge et al., 2021; Mehr et al., 2019; Unyk, Trehub, Trainor, & Schellenberg, 1992). If similar time-dependent songs could be collected in-to a database, it would then be highly interesting to investigate if the audio features of such songs match up with the features that drive the time-of-day preferences uncovered herein. Here, the Spotify API's ability to search user-made playlists for name and description is a highly productive approach, as shown in a recent study uncovering a large amount of variation in sleep music (Scarratt, Heggli, Vuust, & Jespersen, 2021).”

Minor comments:

p8, line 52 “...clusters exhibited a consistently sequential relationship across the week”

If I understood correctly, the k-means clustering was just on the audio features and not any temporal features? It would be good to state this, if so.

That is correct, the clustering was only performed on audio features, and not any temporal features. However, each row in the data correspond to one specific hour of the week, so by inspecting the cluster labels of the rows we were able to see the sequential pattern repeating across the week.

We have now updated the paragraph to more clearly convey this.

p.10, line 47

Could you elaborate on what exactly was conceptually replicated here? I.e., is it just that there are diurnal patterns generally, or that the specific features shift in similar ways?

Absolutely! We have now amended the sentence to clarify that we conceptually replicate the general presence of diurnal patterns in music consumption, but not necessarily patterns in specific features. The previous study that lies closest to ours is the Park et. al. 2019 study where they used a similar range of audio features. However, as they performed data reduction on the audio features resulting in highly complex compound measures, they do not easily map on to our raw data, and we hence refrain from claiming a direct replication on particular features.

p. 11, line 35

The examples of specific songs are nice, however, it would be good to include (perhaps in brackets) a brief elaboration of what is normal or extreme about the audio features in these songs.

Thank you for the suggestion, which we have implemented. We describe Every Breath You Take as a *“mid-tempo soft rock ballad”* and Svefn-g-englar as a *“slow, ambient, and dreamy song”*.

In all, this was an interesting paper and thanks to the authors for a fun read.

Thank you!

Responses to Reviewer 2

The manuscript presents an interesting study on the variance of music consumption behavior as reflected in Spotify listening histories over each hour of the week. To this end, the authors leverage the MSSD. While most results are not overly surprising, the work contributes to research on data analysis carried out in the fields of music information retrieval and music recommender systems.

The paper is generally well-written, language-wise. However, several parts are quite confusing. Some of this confusion certainly originates from the fact that the results are presented before the actual methodology is introduced (something I am frankly not used to, even though I already reviewed hundreds of scientific papers).

Nevertheless, the authors should do a better job in clarifying important methodological details earlier in the manuscript. Also, they should provide already in the introduction a very clear statement of their actual contributions and individual experiments carried out. For instance, when reading the manuscript, it was very unclear to me in which experiments which kind of data (or which datasets) were used, in particular, when and why some experiments used individual tracks and some used "playlists". I first thought the authors confused "playlists" with "listening sessions", which are very different things; because MSSD contains only listening sessions, not hand-crafted playlists. Only later it became somewhat clear that the authors additionally gathered users' playlists and used them for additional experiments/user studies. All in all, it is hard to understand which experiments use MSSD, which rely on Spotify users' individual user-generated content, and which rely on data from other sources. All this confusion could have been avoided by clearly mentioning in the introduction explicit research questions and main methodological aspects (e.g., statistical data analysis for the large-scale data-driven study, user study involving analysis of Spotify users' own playlists, etc.).

Thank you for this comment. We have now worked through the manuscript to ensure a better separation between the individual experiments and clarified the distinction between listening session, playlist, and individual tracks.

One part of the manuscript particularly suited to exemplify this problem is Section 2.4 "Playlists, but not individual tracks, reflect diurnal musical preference". From the text provided in this section, it does not even become clear how and why *playlists* were analyzed. In fact, the text in this whole section does not even mention the word "playlist" again. Therefore, from the running text in Section 2.4, I do not see any empirical evidence for the author's statement made in the section name. Table 1 is no more informative either. E.g., "Mean" numbers are reported, but what the scale is (on which values are these means computed?) remains unclear.

The headline of Section 2.4 was intended to highlight that we did not find a clear-cut diurnal preference for individual tracks, as opposed to the clear preference for playlists discussed in the previous section. To address this issue, we have now changed the headline of Section 2.3 to *"Awareness of diurnal musical preference in playlists"* and the headline of Section 2.4 to *"No clear diurnal musical preference for individual tracks"*.

We have also amended Table 1 and the table text to better convey that what was previously listed as "Mean" is the mean preference for listening to a given track at its predicted

subdivision of the day. As is explained in detail in the methods section, this value is calculated by taking the participant's rating (on a continuous slider ranging from 1 to 101) for a given representative track of one subdivision of the day, and then subtracting the mean of the participant's rating of all the other subdivisions of the day.

I very much appreciate the authors' discussion of limitations, in particular, the inherent data "bias towards western culture" and the inability "to investigate factors such as age". However, both could be resolved to some extent by repeating the statistical analysis on additional datasets such as LFM-1b or MLHD (for details, see my comments below in "Additional remarks and suggestions").

Thank you! We very much look forward to continuing this work, especially with linking our findings to the multiple datasets that are based on data provided from last.fm, Reddit, and Twitter. In regards to the LFM-1b, LFM-2b, MLHD, and a few other such datasets, these are rich sources for demographic information as well as general music listening habits. However, many of these does not contain any audio-based information, such as the audio features which are the main focus of our present study. As such, it is not presently possible to repeat our statistical analysis on these datasets. Doing so would require matching tracks in those datasets to tracks with pre-computed audio features available through the Spotify API, a task which is beyond the scope of the current study. That said, we are in the process of setting up pipelines calculating basic audio features using the MIRtoolbox, which will eventually allow us to perform correlations with the proprietary features available from Spotify and in turn generalize to other datasets. However, given the current implementations of audio feature calculations, such a pipeline would take 2.3 years of compute time to cover just the MSSD. We are currently working on improvements and hope to publish this pipeline in the near future.

Regarding the study on playlists, I am wondering why the authors did not consider the standardized Million Playlist Dataset (also by Spotify), see: <https://www.aicrowd.com/challenges/spotify-million-playlist-dataset-challenge>.

Thank you for suggesting this dataset, which we hope to take a look at in future studies. However, for the current study this particular dataset was publicly re-released after our project concluded (September 2020), and we found it challenging to use in its current state due to Spotify's statement that the playlists included have been both manually filtered for both offensive content and quality, and have fictitious tracks added to them.

Related to the features under investigation, the authors rely on Spotify's audio features even though they are intransparent and their computation is a black box (which authors honestly admit). I, therefore, think that authors should also consider and investigate semantically more meaningful features instead of barely graspable concepts of "organism" or "bounciness". What comes to mind quickly is genre information (which is provided by Spotify too).

This is a good suggestion. Genre information is indeed a highly interesting data point, especially since it is a categorization that often covers more than just the sound of the music, by placing a particular piece of music as belonging to tradition, or within a set of conventions. However, there are two main reasons we did not investigate genre in our current study. The first is technical. The MSSD does not provide genre tags, and the track IDs are scrambled using an unreversible hashing algorithm. This makes it impossible to get genre information for the tracks in our main analysis. Secondly, Spotify appears to be using a wide

range of genre tags covering at least 5521 different tags covering both broad classifications such as “heavy metal” and highly specific tags such as “Wisconsin indie” (for a full list see <https://everynoise.com/>). In addition, these tags are artist-based and will therefore not discriminate between a lullaby and dance song. We therefore focused our project on the audio-dependent features provided by Spotify.

Additional remarks and suggestions:

- The authors claim that MSSD is the "to-date largest publicly available dataset on music listening behaviour including information on when, during the day, a particular track is listened to". I am not sure that this statement is true. Please also consider mentioning the MLHD

([https://ddmal.music.mcgill.ca/research/The Music Listening Histories Dataset \(MLHD\)/](https://ddmal.music.mcgill.ca/research/The_Music_Listening_Histories_Dataset_(MLHD)/)) and the LFM-1b and LFM-2b datasets (<http://www.cp.jku.at/datasets/LFM-1b/> and <http://www.cp.jku.at/datasets/LFM-2b/>).

Thank you for pointing out these interesting datasets, which we now cite in the introduction.

- A screenshot of the UI used in the user studies would be highly appreciated to support the textual explanations.

We have now made the code used for running the playlist classification study available on GitHub: <https://github.com/OleAd/SpotifyPlaylistClassifierApp>. This a Node.js-application which originally ran on Google App Engine, but will also run locally.

- Details about the survey participants should be given: how were they recruited? which platform was used (Amazon MTurk)? are there demographic biases?

Participants for the survey was recruited organically through email lists and Twitter. In order to only collect fully anonymous data, we did not collect any demographic information, but we are likely to see a similar bias as discussed in the manuscript.

- Please include in each figure (and/or caption) a very explicit indication about which values are shown on the axes (in particular the y-axes). For instance, Figure 1A just says "Audio features" on the y-axis with a numeric range between -3 and 3, which is unclear (because most Spotify features have a range between 0 and 1). The same issue can be seen for Figure 1B, and others.

Also include, in Figure 1A, a legend telling the reader which color represents which audio feature.

We have now updated the figures to include clearer axis labels, as well as added a legend to Figure 1A.

- It would be very interesting to include a visualization similar to Figure 2 but showing working days vs. weekends.

We agree that the difference between working days and weekends is highly interesting. The cluster centroid values in Figure 2 are the result of the clustering algorithm settling on the per-day subdivisions, and cannot be extended to show working days vs the weekend in its

current state.

- "This indicates that people, in general, listen to a wide variety of music, as evident by the small changes in mean audio feature values between subdivisions.": Why is a small change in the audio features' *mean* evidence for a wide variety? A *large standard deviation* would rather provide such evidence, in my opinion.

We agree, and intended to communicate that both the mean and the standard deviation comes into play in this claim. We separate address the standard deviations in the subsequent paragraph, and have for clarity amended the sentence to read:

"This indicates that people in general listen to a wide variety of music, as evident by the small changes in mean audio features values between subdivisions and uniformly large standard deviations (for an overview see Supplementary Table 1 and Supplementary Figure 2)."

- "absolute audio feature values only exhibit qualitatively small changes. For instance, the mean Tempo lies in the range of 122.3 to 122.8 BPM": I guess "qualitatively" should read "quantitatively" here.

We have now changed the wording to *"perceptually small changes"* for improved clarity.

- "to ensure a wider geographical and cultural representation. Collating such datasets would require collaboration with the music streaming industry...": There do exist datasets that cover users with a wide variety of cultural backgrounds/different countries. I suggest having a look at the LFM-1b, LFM-2b, and MLHD, for instance (see above).

Please see previous reply.

- Please add in Section 4.1 a link to the MSSD.

Link now added.

- Please clarify the undefined terms and unclear choices you used/made throughout the manuscript (e.g., in Sec 2.1, what is an "optimal division" (optimal in which sense? according to which measure/metric? how do you guarantee optimality?); "playlist" versus "(listening) session" should be clearly defined and made clear in which experiment each is used (see my comment above); Section 4.5: "the network takes six audio features" => why only 6 and why those 6?

We have worked through the manuscript to better separate between playlists, listening session, and individual tracks.

Sec. 2.1: In response to previous comments, we have amended the two sentences starting section 2.1, as well as made the clustering process clearer in the methods section 4.1.

Sec. 4.5: We have now amended the sentence to refer back to the previous two paragraphs explaining the selection of these six audio features:

“The network takes six audio features (danceability, energy, loudness, liveness, valence, and tempo, due to the highly non-normal distributions of speechiness, acousticness, and instrumentalness) as inputs into five fully-connected hidden (...)”

- In Sec 4.7, the difference between proportions of adjacent activities in the ranking is expressed in positive numbers first ("0.1"), but then it's said: "The resulting value lies in the range 0 to -1." Please be consistent.

Thank you for noticing this error, we have added a minus sign before the first number.

- Correct a few grammar mistakes: "audio features relates to" => "relate to"; "Red indicate a relative" => "indicates"; "analyzed music tracks that was" => "were"

Thank you, fixed!

- From the (rather vague) description of the approach in Sec 4.1 (e.g., using terms such as "the data" without saying exactly which data/features), I do not get why the "five clusters identified by the k-means clustering exhibited a sequential relationship across the 168 hours". Was the sequential characteristic considered when computing k-means? Was one clustering for each of the 168-hour-bins computed and clusters tracked over time? If so, how were they tracked? If not, how can you obtain a sequential relationship from the single clustering of all the data?

The k-means clustering calculates clusters in the data, and thereafter assigns a cluster label to each row of unique data (in this case the unique hours of the week). Hence, we were able to see that the clusters exhibited a sequential relationship. We have now amended Section 4.1, making it clearer that we cluster on audio features per unique hour of the week, added a formula for the normalization function, and improved the clarity of the clustering process.

- Instead of saying that the StandardScaler function of scikit-learn was used, rather provide a formula of what it does.

We have now provided a formula and explanation:

“This function normalizes data by removing the mean and scaling to unit variance, such that the standard score z of sample x is calculated as $z = (x - u)/s$, wherein u is the mean and s is the standard deviation.”

- Talking about formulas, I would highly appreciate a formula instead of (or in addition to) the lengthy description of "circular error distance" at the end of Sec 4.5.

Absolutely! We have now provided a formula:

“Here the error value is the shortest step-wise distance between the participant’s response and the network’s classification as given in the following equation: $CircErr = \operatorname{argmin}(res - pred, 5 + pred - res)$ where res is the participant’s response and $pred$ is the network’s classification both in the range $[1,5]$.”

Appendix B

Dear Professor Joydeep Bhattacharya and Professor Essi Viding

We have now worked through the last round of useful and constructive suggestions from the reviewers, and believe our manuscript has improved substantially from the initial submission. In particular, we have made a small change to font sizes on Figure 1, ensured proper citation style of conference proceedings, and made minor adjustments to a number of sentences following the suggestions from the reviewers.

We hope that our revisions and detailed replies to both reviewers (see below) are satisfactory, and look forward to seeing the manuscript published.

With kind regards on behalf of the authors,
Ole Adrian Heggli

Response to reviewer 3

The authors have addressed the key issues we raised in our initial review, and we both think it is mostly ready for publication.

Below are a few additional points the authors may wish to incorporate (all page numbers are to the pages of the proof document, e.g., out of 63 pages):

- p18 line23 - The sentence "In a broader perspective..." seems a bit tacked on, and feels like it either needs more elaboration, or to be removed.

We agree, and have removed the sentence.

- You can probably better integrate your new text on page 20, line48-51 and the following section, since you say that the MSSD contains pre-calculated musical audio features two sentences in a row.

We agree, and have amended the second sentence.

- On another read of the whole paper, one thing that stood out was that the jump from the introduction to the results section felt like it needed more preparation. I.e., you introduced the fact that diurnal patterns occur in nature and human behaviour, and music specifically, and then say you are going to use the MSSD to investigate the latter, and briefly describe this dataset... and then you jump straight into the analysis. It felt like it needed 2-3 more sentences to reiterate what exactly it is that we do not know and stand to gain from the study. E.g., the final couple sentences were basically "Prior research... has shown..." and it felt like it needed something like "But, it is not clear whether..." and a "We now use the MSSD, along with X computational methods to provide some answers to these questions".

Thank you for this suggestion. To make the jump to the results a bit clearer we have now changed that last sentence to read: "By leveraging the rich information in the MSSD along with two behavioural follow-up studies we show that audio features allow us to quantify distinct diurnal patterns of musical content."

- The y-axis text in Fig.1b is small and hard to read

We've increased the font size.

- page 24, line 18 - Here you add a sentence in light of our previous comment. This helps, but actually I would recommend both being slightly more direct AND would bring this up at the start of the paragraph. E.g., from the start of the paragraph "In Figure 2 we show... . While these mean differences are themselves too small to be perceived (see section 2.2), here, they serve as indicators of underlying trends. We found...".

Thank you for this suggestion. We have moved and amended the sentence so that it appears at the start of the paragraph: "In Figure 2 we show how the subdivisions' audio features relate to the individual audio feature's grand mean. While these values are small in terms of musically perceivable differences (see section 2.2) they serve as useful indicators of underlying trends. We found (...)"

- page 13, line 48 - You conjecture that a song with characteristics close to mean would be common across times of day, whereas songs with characteristics closer to the extremes in feature-space would be more likely unique to single distributions... this seems intuitive, but perhaps not trivially true. I assume you did not run any analyses on this? This section could benefit from even a simple analysis to at least partly backup this claim empirically, or at least clarify that this is a speculation and not a finding from your analyses.

We have now clarified that we are speculating, as we have not run any supplementary analyses: "A possible interpretation here is that a song with audio features falling close to the mean values for a given subdivision is also likely to be found in another subdivision. However, a song exhibiting audio features close to the extreme of the distribution in a given subdivision is less likely to be found in another subdivision. In other words, we conjecture that whereas *Every Breath You Take* by The Police (a mid-tempo soft rock ballad) may be listened to uniformly throughout the day, *Svefn-g-englar* by Sigur Rós (a slow, ambient, and dreamy song) may trend during the Night."

Again thanks to the authors, and all the best for the paper.

Thank you!

Response to Reviewer: 2

The revised manuscript has improved much over the original version. Authors have addressed my comments in their response, mostly in a satisfactory manner, and updated their manuscript accordingly.

I also have to clarify that, given my background in computer and data science, I reviewed the manuscript of course taking such a perspective, acknowledging that expectations and requirements in the medical and psychological disciplines might be different (for instance, most researchers in CS/DS/AI would complain about the use of the very simple k-means clustering algorithm when nowadays ample alternatives exist). From this point of view, I found a bit strange what the authors say in their response to justify why they did not use other datasets to investigate whether results generalize. While it is true that LFM-* and MLHD do not come with audio features, matching the tracks to Spotify URIs and fetching the respective audio features from the corresponding Spotify API endpoints certainly does not take years. A Master's student of mine managed to do this in a couple of days.

Notwithstanding, this is (still) a very interesting paper with some novel findings!

Thank you! We acknowledge our limited experience in the CS/DS/AI-field and hope to improve in the future.

On the other hand, there are still a few issues that need to be addressed before acceptance:

- I assume there will be a final copyediting step. When referencing papers that appeared in conference proceedings, authors currently use an incomplete format, in particular, they mention "editors", but don't give the names of those editors (just the author names). They also don't give the corresponding pages of the paper in the proceedings. Furthermore, some conferences are abbreviated, while others are not: e.g., "27. Vigliensoni G, Fujinaga I, editors. The Music Listening Histories Dataset. ISMIR; 2017." vs. "28. Schedl M, editor The lfm-1b dataset for music retrieval and recommendation. Proceedings of the 2016 ACM on international conference on multimedia retrieval; 2016.". All of this is in stark contrast to the complete and seemingly correct format authors use for journal articles.

Thank you for noticing. We found that this occurred as a result of an incomplete formatting style in our referencing software, which have now been fixed.

- The manuscript should be checked for consistency in US vs. UK spelling.

Checked.

- On page 20/63, authors now mention the LFM-2b dataset, but do not include the corresponding reference:

Alessandro B. Melchiorre, Navid Rekabsaz, Emilia Parada-Cabaleiro, Stefan Brandl, Oleg Lesota, Markus Schedl: Investigating gender fairness of recommendation algorithms in the music domain. *Inf. Process. Manag.* 58(5): 102666 (2021)

Thank you for providing this reference, which was not published when our revision was submitted. It is now inserted in the correct place.

- On page 30/63, authors should also report the mean accuracy achieved by their NN over all classes (they do so only for each class separately, and only the mean error is provided overall).

The participant agreed with the NN's predicted subdivision for 63% of the playlists, as stated in the opening sentence of the paragraph. The NN's accuracy on holdout data (97.16%) is reported in the Methods, section 4.5.

- On the same page, when discussing the results of the NN experiments, I suggest to be a bit more cautious. For instance, I don't think it's fair to say that results indicate that "the participants in general agreed with predicted subdivisions". From the results it seems they rather "were more likely to agree with the prediction of the classifier than with a random prediction" (a dumb baseline classifier that randomly picks a class).

Thank you for this suggestion, which we've incorporated: "For all subdivisions the predicted classification had the highest level of agreement, indicating that the participants were more likely to agree with the prediction of the classifier than with a random prediction (...)"

- Again here, "These results suggest that people are ... to a high degree in agreement about which type of playlists they prefer". This sentence can be confusing as it might imply that people are in agreement with each other, not with the predictions. I suggest to add "in agreement with the predictions".

Agreed, we've changed the sentence: "These results suggest that people are consciously aware, opinionated, and in agreement with the predictions about which type of playlists they prefer for the different subdivisions of the day."

- On page 33/63, "...both in general listening habits and in user-created playlists, we wanted to see if these preferences exist also in individual tracks". When reading "general listening habits" I first thought that this means computed on individual tracks. But that's what authors set up to investigate next. I suggest to be more precise and indicate (again) how these "general listening habits" are defined/computed.

We have now amended the sentence to read "both in the general listening habits contained within the MSSD and in user-created playlists (...)".

- Page 36/63: "... patterns ... reflect*s*"

Fixed.

- Page 42/63: Authors should add a reference to the MIR toolbox, at least to the webpage: <https://www.jyu.fi/hytk/fi/laitokset/mutku/en/research/materials/mirtoolbox/> (I think there is a paper reference, too).

A reference to the MIR toolbox is already present, in citation 31.

- On page 47/63, authors say that the network contains five fully-connected hidden layers. However, authors report just four values for the number of units in each layer. Looking at Figure 3, indeed, it seems that there are just 4 hidden layers + 1 input + 1 output layer.

Thank you for spotting this. Changed to four.

- On page 48/63, the UI used in the user study is described in pretty much detail. I still think a screenshot should be given here to better understand the look&feel of the UI. At the very least, authors should explicitly refer to the additional material and add a link to screenshots of the UI.

We now explicitly refer to the repository at the end of the section.